# Physiological and pathophysiological control of synaptic GluN2B-NMDA receptors by the C-terminal domain of amyloid precursor protein

Paula A Pousinha[1]*, Xavier Mouska[1], Elisabeth F Raymond[1], Carole Gwizdek[2], Gihen Dhib[1], Gwenola Poupon[2], Laure-Emmanuelle Zaragosi[3], Camilla Giudici[4], Ingrid Bethus[1], Emilie Pacary[5], Michael Willem[4], Hélène Marie[1]*

[1]Team Molecular Mechanisms of neuronal plasticity in health and disease, Institut de Pharmacologie Moléculaire et Cellulaire, Centre National de la Recherche Scientifique, Université de Nice Sophia Antipolis, Nice, France; [2]Team SUMOylation in neuronal function and dysfunction, Institut de Pharmacologie Moléculaire et Cellulaire, Centre National de la Recherche Scientifique, Université de Nice Sophia Antipolis, Nice, France; [3]Team Physiological genomics of the eukaryotes, Institut de Pharmacologie Moléculaire et Cellulaire, Centre National de la Recherche Scientifique, Université de Nice Sophia Antipolis, Nice, France; [4]Ludwig-Maximilians-University Munich, Munich, Germany; [5]INSERM U1215, Neurocentre Magendie, France et Université de Bordeaux, Bordeaux, France

*For correspondence: pousinha@ipmc.cnrs.fr (PAP); marie@ipmc.cnrs.fr (HM)

**Competing interests:** The authors declare that no competing interests exist.

**Abstract** The amyloid precursor protein (APP) harbors physiological roles at synapses and is central to Alzheimer's disease (AD) pathogenesis. Evidence suggests that APP intracellular domain (AICD) could regulate synapse function, but the underlying molecular mechanisms remain unknown. We addressed AICD actions at synapses, per se, combining in vivo AICD expression, ex vivo AICD delivery or APP knock-down by in utero electroporation of shRNAs with whole-cell electrophysiology. We report a critical physiological role of AICD in controlling GluN2B-containing NMDA receptors (NMDARs) at immature excitatory synapses, via a transcription-dependent mechanism. We further show that AICD increase in mature neurons, as reported in AD, alters synaptic NMDAR composition to an immature-like GluN2B-rich profile. This disrupts synaptic signal integration, via over-activation of SK channels, and synapse plasticity, phenotypes rescued by GluN2B antagonism. We provide a new physiological role for AICD, which becomes pathological upon AICD increase in mature neurons. Thus, AICD could contribute to AD synaptic failure.

## Introduction

Human genetic evidence indicates that the amyloid precursor protein (APP) plays an important physiological role in the central nervous system and is central to the pathogenesis of Alzheimer's disease (AD). In humans, a polymorphism in APP that reduces APP processing protects from sporadic Alzheimer's disease (AD) and normal aging-dependent decline (*Supplementary file 1*; *Jonsson et al., 2012*). In contrast, mutations in APP and in genes that regulate APP processing are causative of AD pathology (*van der Kant and Goldstein, 2015*). APP is a transmembrane protein that undergoes extracellular cleavage by one of two activities, α- or β-secretase, resulting in the formation of large N-terminal extracellular fragments of secreted APP and smaller, membrane-bound C-terminal fragments. Subsequently, the membrane-retained C-terminal fragments are subjected to an

intramembranous scission by the γ-secretase complex (*Wolfe et al., 1999*) to generate the APP intracellular domain (AICD) and simultaneously release p3 (after α-secretase cleavage) or amyloid-β peptide (Aβ, after β-secretase cleavage). In collaboration with other teams, we also recently described a new secretase activity, called η-secretase (*Willem et al., 2015*). Cleavage by η–secretase occurs primarily at amino acids 504–505 of APP695 and releases a ~ 500 amino acid-long ectodomain and a membrane-bound fragment called CTF-η, which is further processed by α- and β-secretases to release a long and a short Aη peptide (Aη-α and Aη-β, respectively). Importantly, APP processing is regulated by synaptic activity (*Kamenetz et al., 2003*; *Cirrito et al., 2005*) and one of the early neurochemical changes in AD is the dysfunction of cholinergic and glutamatergic synapses, which correlates with cognitive decline ante mortem (*Selkoe, 2002*). It is therefore necessary to elucidate the physiological role of APP at the synapse by clarifying which functions are mediated by APP itself and by its proteolytic cleavage products. These data will be crucial to better understand how changes in APP expression/ proteolytic processing contribute to synapse failure in AD pathogenesis (*Selkoe, 2002*).

When studying APP function, a confounding factor is the presence of the functionally redundant APP-like protein-1 and -2 (APLP1 and APLP2), which display strong similarity to APP within the cytoplasmic domain. APLP1 and APLP2 are processed like APP (*Müller and Zheng, 2012*) and release intracellular peptides that, like AICD, can potentially regulate gene transcription (*Scheinfeld et al., 2002*). Recently, it has been shown that the APP intracellular domain might play a crucial role on synapse physiology, given that mice lacking the last 15 amino acids of APP (APP-CT15 domain) in an APLP2-KO background (APPΔCT15-DMs) present hippocampal synaptic deficits. In particular, these mice showed impaired long-term potentiation (LTP), apparently due to post-synaptic alterations, and deficits in spatial working memory (*Klevanski et al., 2015*). Yet, the synaptic mechanisms associated to these synaptic and behavioral alterations have not been clarified. AICD is very short (~50 amino acids), but includes important sequence motifs and adapter protein-binding sites (*Grimm et al., 2015*), raising the possibility that APP functions could be dictated by molecular interactions of AICD with cytoplasmic proteins at the membrane or after its release from the membrane by γ-secretase cleavage. This 6 kDa peptide is unstable, hence difficult to isolate and characterize biochemically (*Belyaev et al., 2010*; *Flammang et al., 2012*). However, studies on cells, in vitro, demonstrated that cleavage of APP in endosomes, through BACE1-dependent amyloidogenic processing, generates an AICD fragment capable of inducing nuclear signaling (*Goodger et al., 2009*; *Grimm et al., 2015*; *Kimberly et al., 2001*). In line with these findings, AICD was shown to shuttle to the nucleus, via binding of the highly conserved YENPTY motif to the nuclear adaptor protein Fe65, but whether AICD harbors transcriptional activity remains controversial (*Cao and Südhof, 2001*, *Cao and Südhof, 2004*; *Riese et al., 2013*).

The fact that the APPΔCT15-DMs present synaptic deficits and that the AICD domain interacts with multiple cytoplasmic partners, presenting bioactivity, suggests that APP, via AICD, might regulate synapse function. To specifically confirm this hypothesis, we assessed the actions of AICD, per se, by using different technical approaches: in-vivo virus-mediated AICD expression, ex-vivo AICD peptide delivery and APP knock-down by in-vivo electroporation of shRNAs, in combination with whole-cell patch clamp electrophysiology. We now report a critical and previously unrecognized role for AICD on controlling GluN2B-containing NMDA receptors (NMDARs) at CA1 excitatory synapses, via a transcription-dependent mechanism. In addition, we provide strong evidence that the increased presence of AICD in mature CA1 neurons, as previously reported in AD mouse models and human patients, alters their synaptic NMDAR profile to an immature-like phenotype with increased GluN2B-NMDAR contribution. This alteration leads to abnormal signal integration, via over-activation of SK channels, and impaired long-term potentiation (LTP), functional phenotypes that could be rescued by partial GluN2B antagonism.

## Results

### In vivo AICD production in the hippocampus

To investigate the actions of AICD, per se, on hippocampal synaptic function, we used in vivo transduction of neurotropic recombinant adeno-associated viruses (AAV) encoding AICD. This approach enabled us to minimize other confounding factors such as concomitant increase of other bioactive

APP fragments or genetically-induced compensatory effects observed when using transgenic models. The human AICD cDNA sequence was inserted in an AAV vector also expressing GFP (called hereafter AICD virus) (*Figure 1A*). As AICD is unstable but probably harbors transcriptional activity (*Cupers et al., 2001*; *Pardossi-Piquard and Checler, 2012*), we also created an AAV expressing AICD coupled to a nuclear localization signal (NLS) (called hereafter AICD-NLS virus) to potentiate nuclear AICD effects (*Figure 1A*). The AAV vector expressing only GFP was used as control (called hereafter GFP virus) (*Figure 1A*). We optimized sparse transduction of CA1 pyramidal neurons, in vivo, by stereotaxic microinjections of these viruses into hippocampi of 21–22 days old rats. Twelve to twenty days post-injection, acute hippocampal slices were prepared and whole-cell patch-clamp recordings were obtained from CA1 transduced neurons (identified by GFP fluorescence) (*Figure 1B*). To confirm increased AICD production in transduced neurons, we performed single cell

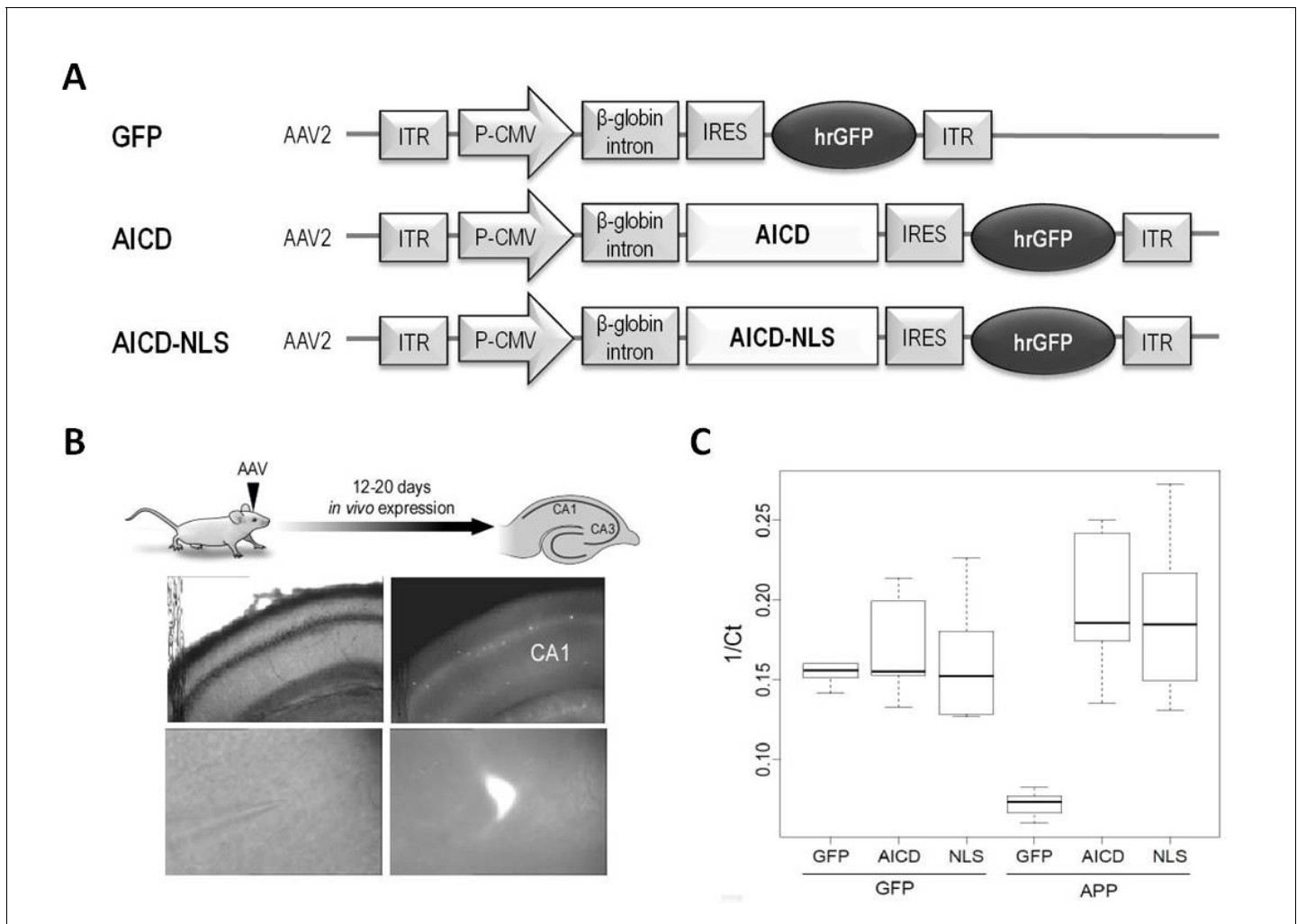

**Figure 1.** Construction and expression of AAVs. (A) AAV constructs: AICD or AICD-NLS were inserted downstream of the CMV promoter and β-globin intron and upstream of an IRES-GFP sequence. ITR: inverted terminal repeats; P-CMV: cytomegalovirus promoter; IRES: internal ribosomal entry site; hrGFP: human recombinant green fluorescent protein. AICD-NES, AICDY682, APLP1–ICD and APLP2-ICD were constructed in this same AAV vector by replacing the AICD cassette with these other cDNAs. (B) Diagram shows schematic of in vivo viral transduction protocol. Photos show low-magnification (x4; top panels) and high-magnification (x60; bottom panels) images of CA1 area of hippocampal slice (left panels show DIC images; right panels show GFP fluorescence) prepared from rat transduced in vivo with AICD-AAV-GFP.(C) Single cell semi-quantitative PCR on cytoplasms of CA1 pyramidal neurons transduced with either GFP, AICD or AICD-NLS viruses. Single cell quantification is reported as 1/Ct and plotted in a box and whisker plot where the horizontal lines represent the median, the box limits delineate the lower and upper quartiles (25% and 75%) and the whiskers are set at minimum and maximum values, for each group. GFP expression was similar on all three groups of cells (left side of panel). Using primers against the C-terminal domain of APP, we detected low levels of mRNA in GFP neurons and higher levels in AICD and AICD-NLS neurons (right side of panel).

quantitative PCR. On average, GFP mRNA expression was similar in GFP, AICD and AICD-NLS neurons (*Figure 1C*; left side of panel). As expected, increased expression of mRNA representing the C-terminal domain of APP was observed in AICD and AICD-NLS neurons (*Figure 1C*; right side of panel).

## AICD increases NMDAR, but not AMPAR, transmission in CA1 pyramidal neurons

To determine whether AICD alters synaptic transmission at the Schaffer collateral to CA1 synapses, we performed whole-cell patch clamp electrophysiology on GFP positive CA1 pyramidal neurons (expressing AICD). We compared the relative contribution of AMPA receptors (AMPARs) and NMDARs to excitatory post-synaptic currents (EPSCs) by measuring AMPAR/NMDAR ratios (*Figure 2A*, *Supplementary file 1*; one-way ANOVA). This ratio was significantly decreased in AICD neurons, an effect that was more pronounced in AICD-NLS neurons (*Figure 2A*), suggesting a nuclear-dependent effect. To confirm this finding, we created an AAV expressing AICD tagged with a nuclear export signal (NES) using the same AAV vector described in *Figure 1A*. Neurons transduced with AICD-NES presented an AMPAR/NMDAR ratio similar to GFP neurons (*Figure 2A*). Phosphorylation of tyrosine 682 in the $^{682}$YENPTY motif within AICD, which regulates binding of cytoplasmic proteins such as Fe65 (*Grimm et al., 2013*), was necessary for this synaptic phenotype. Indeed, we created an AAV expressing AICD with an Y682G mutation (same AAV vector as in *Figure 1A*) and neurons expressing AICDY682G exhibited a normal AMPAR/NMDAR ratio (*Figure 2A*). APP, APLP1 and APLP2 exhibit some functional redundancy and their C-terminal domain is highly conserved (*Shariati and De Strooper, 2013*) (*van der Kant and Goldstein, 2015*). We thus tested if the C-terminal domains of APLP1 and APLP2 could also mediate this phenotype. For this, AAV expressing the last 50 amino acids of APLP1 and APLP2 were created (APLP1–ICD and APLP2-ICD, respectively). Interestingly, APLP2-ICD, but not APLP1-ICD, lowered the AMPAR/NMDAR ratio (*Figure 2A*). Together, these data show that AICD, via its nuclear activity and phosphorylation of its tyrosine 682, affects excitatory post-synaptic transmission, a function also displayed by the APLP2 C-terminal domain, but not that of APLP1.

A change in the AMPAR/NMDAR ratio suggests that a postsynaptic modification in receptor composition and/or number has occurred. It does not indicate whether AMPARs, NMDARs, or both have been modified. To further characterize this phenotype, we first analysed the miniature AMPAR EPSCs (mEPSCs). One could observe similar amplitude and frequency of mEPSCs, suggesting that AICD does not affect AMPAR transmission (*Figure 2B–D*, *Supplementary file 1*, one way ANOVA). We then performed pairwise comparison of AMPAR and NMDAR EPSC amplitudes by double patch clamp of infected and closely adjacent uninfected neurons (*Figure 2E*). We focused on AICD-NLS neurons, which displayed the most significant reduction in the AMPAR/NMDAR ratio. AICD-NLS neurons exhibited unchanged AMPAR EPSCs, confirming the results obtained with AMPAR mEPSC analysis (*Figure 2F and G*; *Supplementary file 1*; matched-pairs Student's t-test), whereas NMDAR EPSCs were markedly increased (*Figure 2H and I*; *Supplementary file 1*, matched-pairs Student's t-test). Together, these data demonstrate that enhanced AICD production in CA1 pyramidal neurons increases NMDAR synaptic transmission, while AMPAR function is not affected.

## AICD regulates NMDAR synaptic currents by increasing synaptic GluN2B contribution, through a transcription-dependent mechanism

In the hippocampus, NMDARs are heteromeric assemblies mainly composed of an obligatory GluN1 subunit and GluN2A or GluN2B subunits (*Rosenmund et al., 1998*). GluN1/GluN2A heterodimers display faster deactivation kinetics than GluN1/GluN2B heteromers, but present similar gating properties (*Vicini et al., 1998*). To investigate if AICD-mediated alterations of NMDAR currents were due to modifications of NMDAR subunit composition, we measured the time constants for the fast, slow and weighted components ($\tau_{fast}$, $\tau_{slow}$ and $\tau_{weighted}$) of NMDAR EPSC deactivation kinetics. AICD-NLS neurons exhibited a slower deactivation time course than GFP neurons (*Figure 3A*), an effect mediated by a significant increase in $\tau_{slow}$ and $\tau_{weighted}$ (*Figure 3A*; *Supplementary file 1*; two-way ANOVA), without noticeable changes in $\tau_{fast}$ (*Figure 3A*; *Supplementary file 1*; two-way ANOVA). These results suggest that AICD causes alterations in NMDAR transmission due to an increase of synaptic GluN2B currents. To test this hypothesis, we evaluated the effect of the selective GluN2B

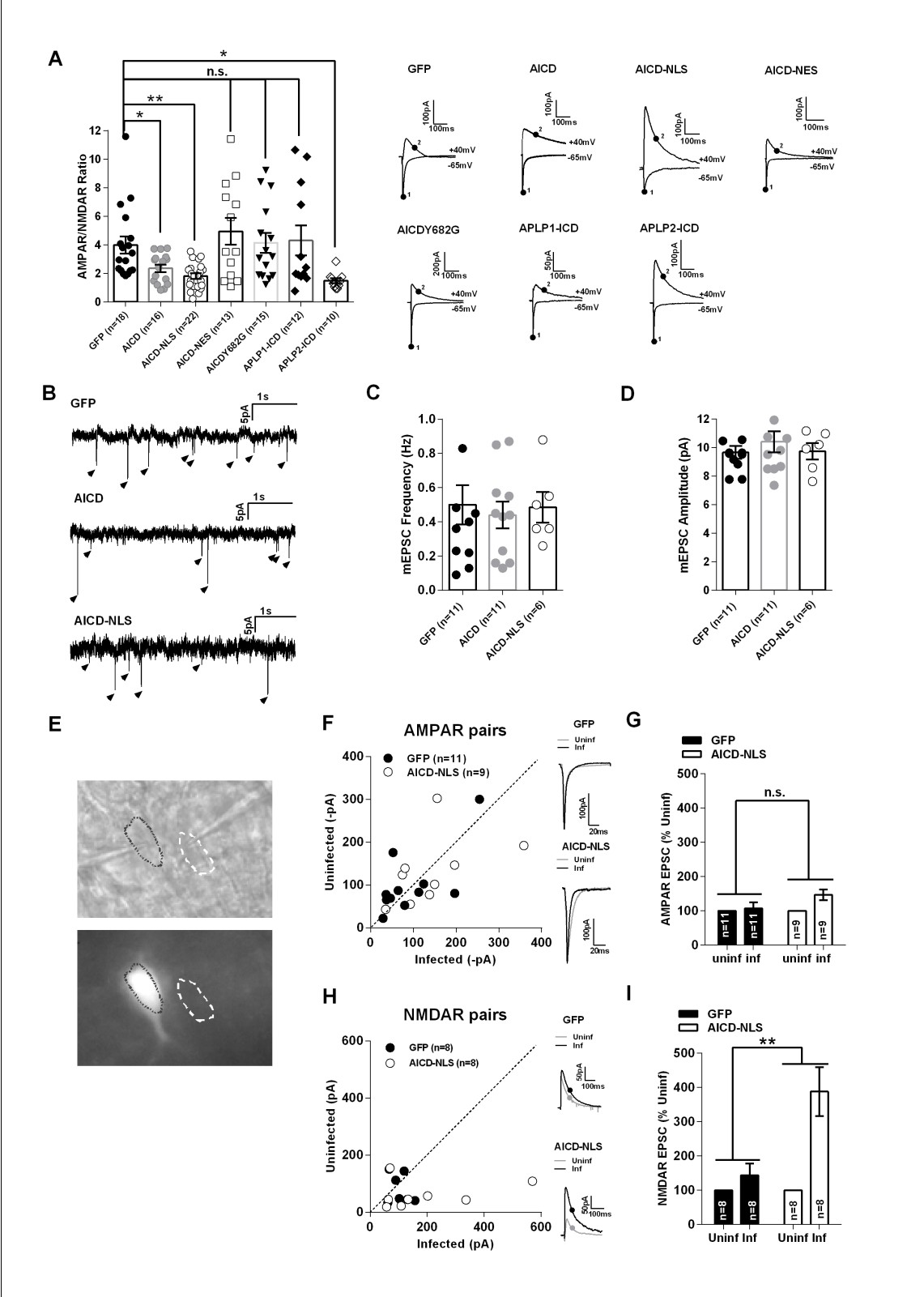

**Figure 2.** AICD increases NMDAR, but not AMPAR, transmission in CA1 pyramidal neurons. (**A**) Scatter dot plot graph shows AMPAR/NMDAR ratios from GFP, AICD, AICD-NLS, AICD-NES, AICDY682G, APLP1-ICD and APLP2-ICD neurons, as indicated. Each dot represents one neuron. Traces show sample AMPAR EPSCs (−60 mV) and NMDAR EPSCs + AMPAR EPSCs (+40 mV) from the different populations of infected neurons. AMPAR/NMDAR ratios were calculated by taking measurements at times indicated (AMPAR, point 1; NMDAR, point 2). (**B**) Example traces of mEPSC recordings from

*Figure 2 continued on next page*

Figure 2 continued

GFP, AICD or AICD-NLS neurons as indicated. (C) Mean frequency, (D) Mean amplitude of AMPAR mEPSCs from GFP, AICD or AICD-NLS neurons. (E) DIC (top panel) and GFP fluorescence (bottom panel) images of a representative example of a pair of infected (AICD-NLS) and non-infected neighbor neurons. (F) Comparison of the amplitude of AMPAR EPSCs from GFP or AICD-NLS neurons with neighboring uninfected neurons. For each pair, the AMPAR EPSC amplitude of the GFP or AICD-NLS neuron is plotted against its uninfected neighbor. Overlays of sample EPSCs from infected (black trace) and uninfected neighboring cells (grey trace) are shown. (G) Average percent change of AMPAR EPSC of GFP and AICD-NLS neurons relative to neighboring uninfected control neurons (normalized to 100%). (H) Same as (F) but for NMDAR EPSCs. (I) Same as (G) but for NMDAR EPSCs. *p<0.05; **p<0.01. Full statistical analysis is detailed in *Supplementary file 1*.

inhibitor, ifenprodil, on NMDAR EPSCs. Ifenprodil (5 µM) significantly reduced NMDAR EPSC amplitude in AICD-NLS neurons, being much less effective in GFP neurons (*Figure 3B,E*; *Supplementary file 1*; one-way ANOVA). Moreover, application of ifenprodil, while not significantly perturbing NMDAR kinetics in GFP neurons, normalized the deactivation kinetics in AICD-NLS neurons (*Figure 3A,B*; *Supplementary file 1*; two-way ANOVA). Enhanced AICD production thus leads to a specific increase in synaptic NMDARs containing GluN2B subunits.

To confirm our finding and exclude putative over-expression-related artifacts of virus mediated in vivo expression, we used an alternative approach to provide ex vivo intracellular delivery of AICD. We synthesized peptides representing AICD-NLS or a negative scrambled control peptide (scrAICD-NLS) linked at the N-terminus to the cell-penetrating domain TAT (*Guscott et al., 2016*). We pre-incubated hippocampal slices with different doses of TAT-AICD-NLS (10, 30 or 100 nM), or vehicle (DMSO; point 0 nM on graph 3D) for at least 2 hr. We then measured the effect of ifenprodil on NMDAR EPSCs amplitude in CA1 pyramidal neurons. We observed a dose-dependent relationship between acute application of TAT-AICD-NLS and levels of synaptic GluN2B currents (i.e. effect of ifenprodil on NMDAR EPSC) with IC50 ~13.88 ± 1.65 nM (*Figure 3D*; *Supplementary file 1*, one-way ANOVA). Notably, doses as low as 30 nM were sufficient to reach maximal incorporation of GluN2B within synapses (~50% of NMDAR current) (*Figure 3D*). This maximal effect was similar to that observed with two weeks in vivo expression of virally encoded AICD and was not observed with incubation of 100 nM of control peptide (scrAICD-NLS; *Figure 3E*; *Supplementary file 1*, one-way ANOVA). Importantly, pre-incubation with TAT-AICD-NES (100 nM) did not significantly increase the contribution of GluN2B-NMDAR to the NMDAR EPSC (*Figure 3E*, *Supplementary file 1*, one-way ANOVA), suggesting that this effect is dependent on AICD presence in the nucleus. To confirm this hypothesis, we tested the effect of ifenprodil on slices pre-incubated in TAT-AICD-NLS together with inhibitors of protein transcription or translation, actinomycin or anisomycin, respectively. As shown in *Figure 3E* (*Supplementary file 1*, one-way ANOVA), TAT-AICD-NLS failed to increase synaptic GluN2B-NMDAR currents in the presence of these inhibitors, confirming that the effect of AICD on synaptic GluN2B-NMDAR is transcription and translation-dependent. In a control experiment, we confirmed that TAT-AICD-NLS is enriched in the nucleus compared to TAT-AICD-NES. As we could not obtain a selective antibody against TAT, we performed this experiment in a mouse neuroblastoma cell line in which APP is knocked out (N2a-APPKO; *Figure 3F,G*) to allow selective identification of exogenously applied TAT-AICD-NLS or TAT-AICD-NES using the APP antibody Y188. As expected (*Figure 3H,I*, *Supplementary file 1*, Student's t-test), TAT-AICD-NLS was enriched in the nucleus compared to TAT-AICD-NES. Together, these data confirm a role of AICD in the regulation of synaptic GluN2B currents via a transcription-dependent mechanism, even at low nanomolar concentrations.

## AICD regulates GluN2B mRNA levels

Our data strongly suggest that AICD could regulate the transcription of *Grin2b*, the gene encoding the GluN2B subunit. To address this possibility, we took advantage of the fact that we had previously collected cytoplasm from transduced neurons with GFP and AICD-NLS (*Figure 1C*). We quantified GluN2B mRNA levels in these samples by single cell quantitative PCR. Unfortunately, our single cell analysis was not sensitive enough to reliably quantify endogenous GluN2B mRNA levels (data not shown). As an alternative, we constructed new high titer AAVs encoding GFP or AICD together with GFP under the synapsin promoter (called hereafter synGFP and synAICD viruses; *Figure 4A*) to direct specific expression of AICD cDNA (without NLS) in neurons with high transduction efficiency

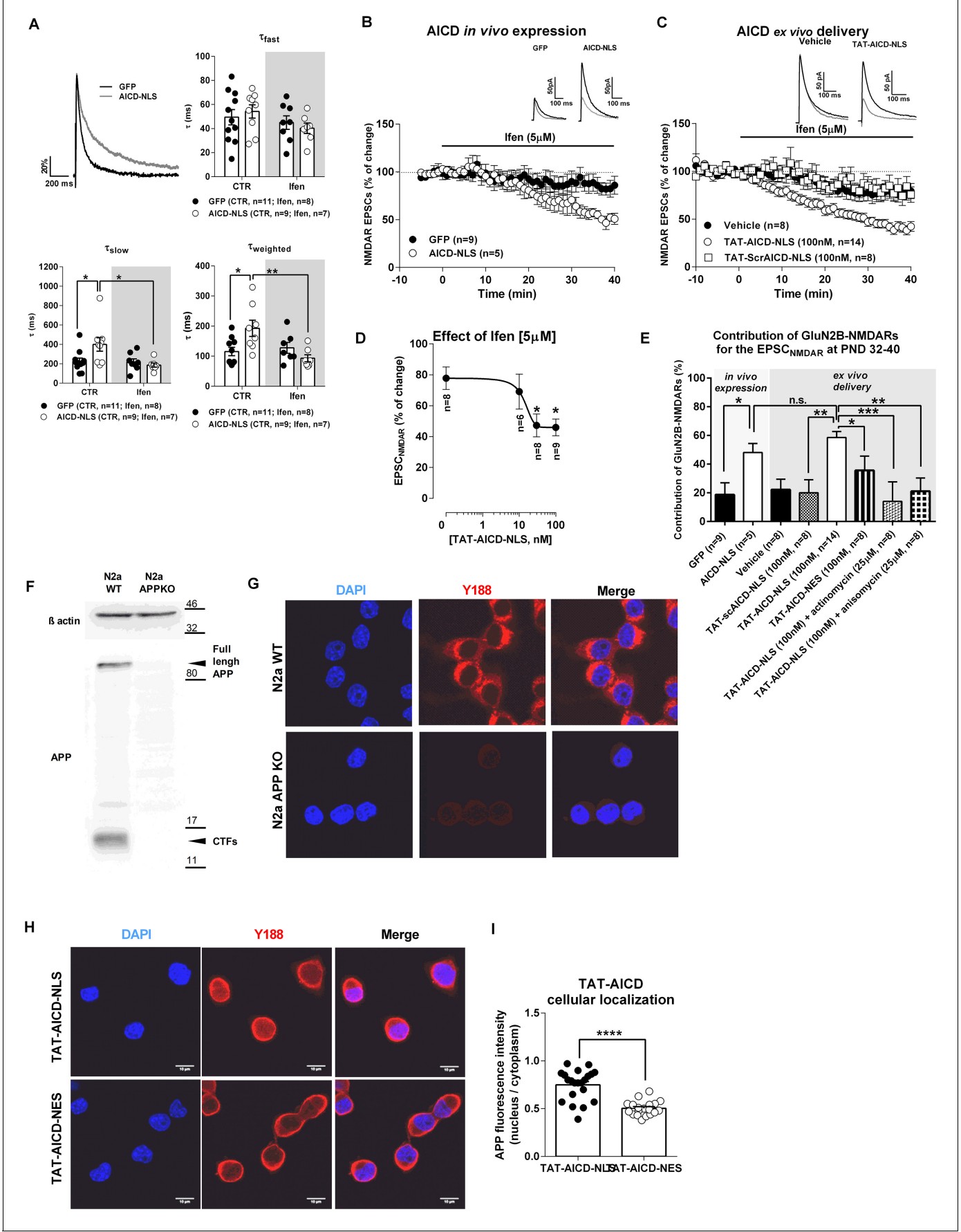

**Figure 3.** AICD increases synaptic GluN2B currents. (**A**) In the upper left corner, we show a comparison of representative whole-cell patch-clamp recordings of pharmacologically-isolated NMDAR EPSCs, normalized to the peak amplitude (in %), from GFP (black line) and AICD-NLS (grey line) neurons, illustrating differences in deactivation kinetics. The other three panels show scatter dot plot graphs with the $\tau_{fast}$, $\tau_{slow}$ and $\tau_{weighted}$ constants, respectively, for NMDAR EPSC deactivation time course in GFP and AICD-NLS neurons before (control; CTR) and after ifenprodil (Ifen; 5 μM) application. (**B**) Time course of ifenprodil effect on NMDAR EPSC amplitude. Traces show sample $EPSC_{NMDAR}$ recorded before (black trace) and after 40 min of Ifenprodil (5 μM) perfusion (grey trace) in GFP and AICD-NLS neurons. (**C**) Time course of ifenprodil effect on NMDAR EPSC amplitude in hippocampal slices incubated with vehicle, TAT-AICD-NLS (100 nM) and TAT-scrAICD-NLS (100 nM). Traces show sample $EPSC_{NMDAR}$ recorded before (black trace) and after 40 min of Ifenprodil (5 μM) perfusion (grey trace) in vehicle and TAT-AICD-NLS neurons. (**D**) Effect of ifenprodil (5 μM) on hippocampal slices pre-incubated with different concentrations of TAT-AICD-NLS (0, 10, 30 and 100 nM). The line represents the nonlinear regression analysis of the data, with IC50 ~13.88 ± 1.65 nM. (**E**) Bars graph showing the contribution of GluN2B-NMDARs for the $EPSC_{NMDAR}$ at PND 32–40 in different conditions, as indicated. (**F**) Representative immunoblot of samples of N2a cells WT or KO for APP (APPKO). APP was detected by antibody against C-terminal domain (Y188) and actin was used as loading control. (**G**) Representative example of N2a-WT and N2a APPKO cells immunostained for APP with Y188. Nuclei were stained with DAPI. (**H**) Representative example of N2a-APPKO cells, pre-incubated with TAT-AICD-NLS or TAT-AICD-NES for one hour were immunostained for APP with Y188. Nuclei were stained with DAPI. (**I**) The ratio of mean APP fluorescence in nucleus/cytoplasm was quantified in N2a-APPKO cells pre-incubated with either TAT-AICD-NLS (n = 55) or TAT-AICD-NES (n = 50). Each dot in B, C, D represents one single neuron. *p<0.05; **p<0.01; ***p<0.001; ****p<0.0001. Full statistical analysis is detailed in *Supplementary file 1* .

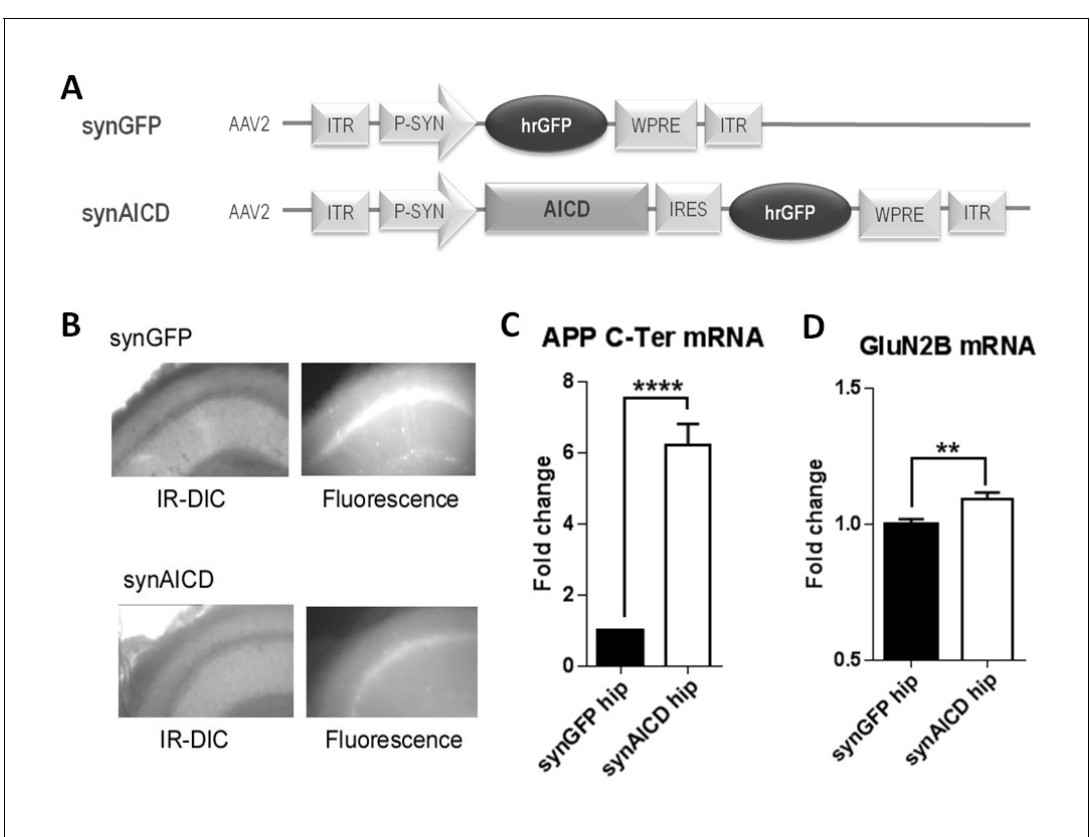

**Figure 4.** AICD regulates GluN2B mRNA levels. (**A**) Diagrams of AAV constructs representing synGFP and synAICD viruses, which exhibit specific expression of AICD in neurons with high transduction efficiencies. AICD was inserted downstream of the synapsin promoter and upstream of the hrGFP sequence via an IRES linker. ITR: inverted terminal repeats; P-SYN: synapsin promoter; IRES: internal ribosomal entry site; hrGFP: human recombinant green fluorescent protein; WPRE: Woodchuck Hepatitis Virus Post-transcriptional Regulatory Element. (**B**) Photos show low-magnification (x4) images of CA1 area of hippocampal slice (left panels show IR-DIC images; right panels show GFP fluorescence) prepared from rat injected in vivo with synGFP and synAICD, as indicated. (**C**) Fold change in APP mRNA expression (detection of C-terminal domain of APP; APP C-Ter) and in (**D**) GluN2B mRNA expression in hippocampi expressing synGFP (GFP hip; 18 hippocampi) or synAICD (AICD hip; 19 hippocampi) as quantified by quantitative PCR (normalization to two house-keeping genes *Rpl13A* and *Ywhaz*). **p<0.01. ****p<0.0001. Full statistical analysis is detailed in *Supplementary file 1*.

(*Figure 4B*). We micro-dissected hippocampi regions highly transduced by synGFP or synAICD for quantitative PCR analysis. As expected, AICD expression, detected with primers against the C-terminal domain of APP (APP C-Ter) was increased in SynAICD-expressing hippocampi compared to synGFP transduced tissue (*Figure 4C*, *Supplementary file 1*, Student's t-test). Importantly, Syn-AICD-expressing hippocampi displayed a significant increase in endogenous GluN2B mRNA levels compared to synGFP-expressing hippocampi (*Figure 4D*, *Supplementary file 1*, Student's t-test), correlating AICD over-expression to increased *Grin2b* expression.

## APP knock-down eliminates synaptic GluN2B NMDARs, an effect rescued by nuclear AICD delivery

Previous work in cultured neurons has shown a link between APP and an increase of synaptic NMDARs (*Cousins et al., 2009*; *Hoe et al., 2009*), but the molecular mechanism remains unknown. Our data indicate that AICD might be the mediator of this APP effect. To confirm this, we decreased APP expression in neuronal precursor cells of the hippocampus, using *in utero* electroporation of shRNAs. We then assessed GluN2B-NMDAR contribution to pharmacologically isolated NMDAR EPSCs during the first week at postnatal development, where most NMDARs at CA1 synapses contain GluN2B (*Monyer et al., 1994*). We aimed to investigate if (1) in vivo APP knock-down could down regulate synaptic GluN2B-NMDARs and if (2) AICD per se could rescue this effect. We used two previously characterized shRNA constructs: shAPP-active and shAPP-inactive (*Young-Pearse et al., 2007*). We first confirmed efficient knock-down of APP expression with shAPP active in a transient transfection assay in hippocampal neurons in culture (*Figure 5A,B*, *Supplementary file 1*, two-way ANOVA). Active or inactive shAPP constructs were *in utero* co-electroporated with GFP plasmid into the hippocampal primordium of E14.5 mouse embryos and mice were sacrificed at post-natal day (PND) 7–9 for experiments. A few hippocampal slices were processed for immunohistochemical quantification of APP levels to confirm decrease of APP expression in vivo (*Figure 5C,D*, *Supplementary file 1*, Man-Whitney test). The other in utero electroporated brains were used to measure NMDARs EPSCs, and effects of ifenprodil, in GFP positive CA1 neurons (*Figure 5E–H*). Inactive shAPP did not affect the synaptic NMDAR function, as ifenprodil effect (~50%) was similar to what has been extensively described for immature neurons (*Figure 5E,H*) (for eg. [*Matta et al., 2011*]). In contrast, ifenprodil was devoid of effect in cells expressing shAPP-active, indicating that loss of APP induces a complete absence of synaptic GluN2B-NMDARs (*Figure 5E,H*; *Supplementary file 1*, one-way ANOVA). Confirming this, neurons expressing shAPP-active exhibited a faster deactivation time course than neurons expressing shAPP-inactive (*Figure 5F*), an effect mediated by a significant decrease in $\tau_{slow}$ and $\tau_{weighted}$ (*Figure 5F*; *Supplementary file 1*; two-way ANOVA), without noticeable changes in $\tau_{fast}$ (*Figure 5F*; *Supplementary file 1*; two-way ANOVA). Notably, ifenprodil only had an effect on these deactivation kinetics in neurons expressing shAPP inactive, while it was ineffective in neurons expressing shAPP active (*Figure 5F*; *Supplementary file 1*; two-way ANOVA).

To test if AICD could rescue, per se, this loss of GluN2B-containing NMDARs at synapses induced by shAPP active, we pre-incubated hippocampal slices PND7-9 pups electroporated with shAPP active with either TAT-AICD-NLS or TAT-scrAICD-NLS, both at 100 nM. Notably, TAT-AICD-NLS fully normalized GluN2B levels at synapses as ifenprodil significantly decreased the amplitude of NMDAR EPSCs (~50%) in neurons expressing shAPP active (*Figure 5G,H*; *Supplementary file 1*, one-way ANOVA), while TAT-scrAICD-NLS had no effect and synaptic GluN2B levels remained low that is, no effect of ifenprodil on NMDAR currents (*Figure 5G,H*; *Supplementary file 1*, one-way ANOVA). Thus, ex vivo delivery of nuclear AICD is sufficient to restore the physiological profile of GluN2B-NMDARs at immature synapses in the absence of APP. To determine if the actions of APP and AICD on GluN2B-NMDARs could be cumulative, in other words, if they could be related to different mechanisms, we investigated if the delivery of nuclear AICD (with TAT-AICD-NLS incubation) could further increase the contribution of GluN2B-NMDARs to NMDAR EPSCs on neurons expressing the control inactive shAPP. As shown (*Figure 5H*; *Supplementary file 1*, one-way ANOVA), pre-incubation with TAT-AICD-NLS does not further enhance synaptic GluN2B-NMDARs levels in these neurons, suggesting that APP acts via AICD to govern synaptic GluN2B-NMDARs levels. Together, these data uncover a crucial physiological role of AICD in controlling GluN2B-containing NMDAR at synapses.

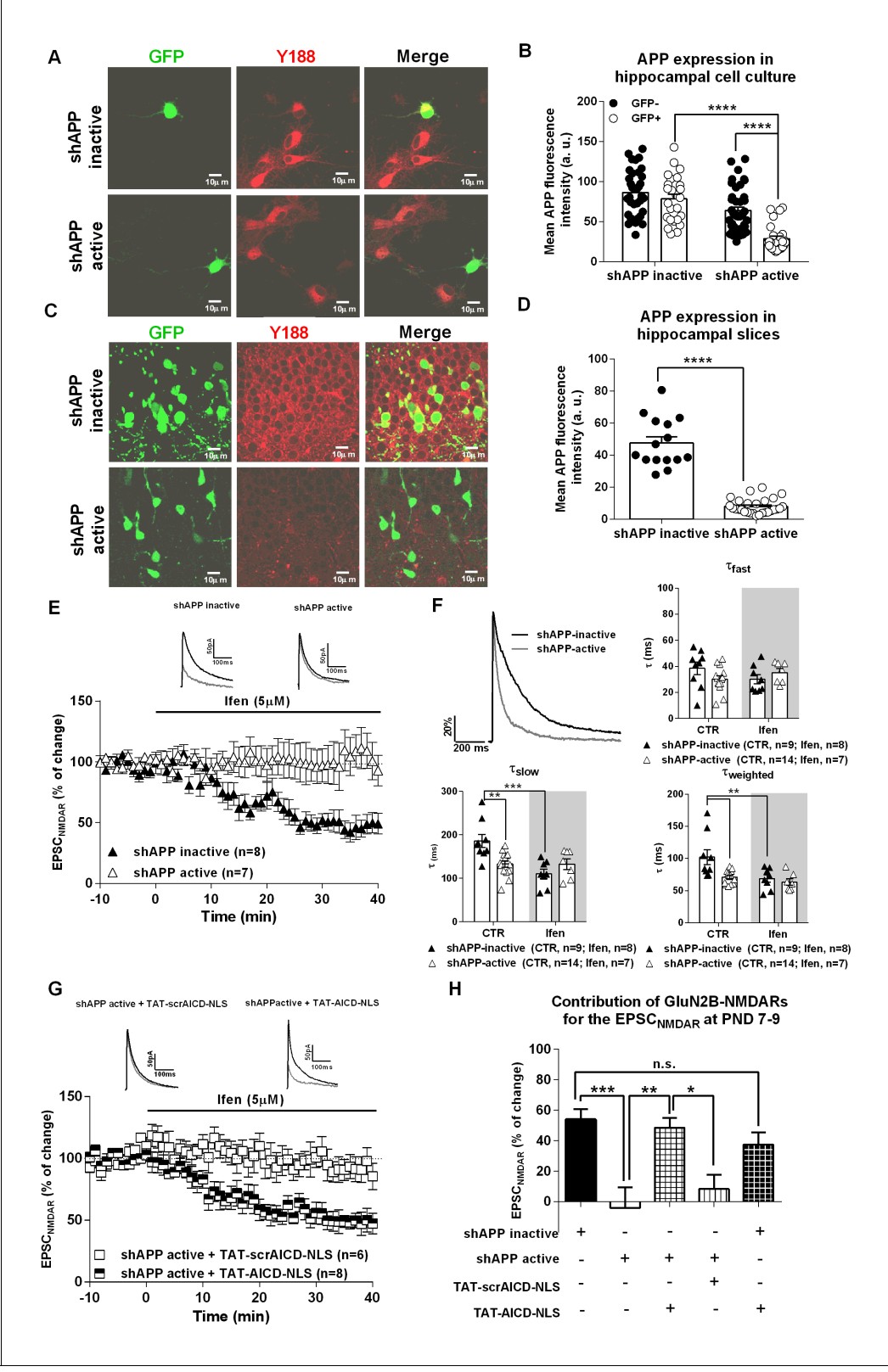

**Figure 5.** APP knock-down eliminates synaptic GluN2B NMDARs, an effect rescued by nuclear AICD delivery. (**A**) Representative examples of hippocampal neurons in culture co-transfected with either shAPP inactive or shAPP active and GFP and immunostained for GFP and APP C-terminal domain (Y188). (**B**) Quantification of APP C-Terminal domain immunostaining in GFP+ (transduced neurons) and neighbor GFP- (non-transduced neurons) in hippocampal neuronal cultures expressing either shAPP inactive or shAPP active shown in (**A**). (**C**) Representative example of hippocampal

*Figure 5 continued on next page*

**Figure 5 continued**

slices of PND 7-9 mice, previously co-electroporated *in utero* with either shAPP inactive + GFP or shAPP active + GFP at E14.5, were immunostained for GFP and APP C-terminal domain (Y188). (D) Quantification of APP C-Terminal domain immunostaining in GFP+ neurons in hippocampal slices after in utero electroporation expressing either shAPP inactive or shAPP active shown in (C). (E) Time course of ifenprodil effect on pharmacologically-isolated NMDAR EPSC amplitude in CA1 pyramidal neurons of PND7-9 mice after being *in utero* electroporated at E14.5 with shAPP inactive or shAPP active, as indicated. Traces show sample $EPSC_{NMDAR}$ recorded before (black trace) and after 40 min of Ifenprodil (5 µM) perfusion (grey trace) in neurons. (F) In the upper left corner, we show a comparison of representative NMDAR EPSCs at PND7-9, normalized to the peak amplitude (in %), in shAPP-inactive (black line) and shAPP-active (grey line) neurons, illustrating differences in deactivation kinetics. The other three panels show scatter dot plot graphs with the $\tau_{fast}$, $\tau_{slow}$ and $\tau_{weighted}$ constants, respectively, for NMDAR EPSC deactivation time course in shAPP-inactive and shAPP-active neurons before (control; CTR) and after ifenprodil (Ifen; 5 µM) application. (G) Time course of ifenprodil effect on NMDAR EPSC amplitude on shAPP active neurons pre-incubated with TAT-AICD-NLS (100 nM) or TAT-scrAICD-NLS (100 nM). Traces show sample $EPSC_{NMDAR}$ recorded before (black trace) and after 40 min of Ifenprodil (5 µM) perfusion (grey trace) in shAPP active/TAT-scrAICD-NLS and shAPP active/TAT-AICD-NLS neurons. (H) Bars graph showing the contribution of GluN2B-NMDARs for the $EPSC_{NMDAR}$ at PND7-9 in different conditions, as indicated. *p<0.05; **p<0.01; ***p<0.001; ****p<0.0001. Full statistical analysis is detailed in *Supplementary file 1*.

## Physiological and pathological levels of AICD in brain tissue

Our data strongly suggest a dynamic correlation between the intracellular levels of AICD and synaptic GluN2B-NMDARs function in both immature and mature synapses. Importantly, we could demonstrate that low nanomolar concentrations of TAT-AICD-NLS are enough to trigger this effect. We then asked if the amount of exogenous AICD, delivered to neurons upon 2 hr incubation of brain slices in TAT-AICD-NLS, is comparable to AICD physiological levels and if its addition to endogenous AICD levels could reach the level observed in an AD mouse model. To access the amount of TAT-AICD-NLS incorporated in the hippocampal cells, we pre-incubated hippocampal slices with TAT-AICD-NLS (100 nM) for two hours and then processed this tissue for western-blotting analysis. In each experiment, the TAT-AICD-NLS mass was calculated based on a dose-response calibration curve of known doses of TAT-AICD-NLS (*Figure 6A*). As shown (*Figure 6C*), the amount of TAT-AICD-NLS incorporated in hippocampal slices after 2 hr incubation was ~10 pg per µg of tissue lysate, which is approximately twofold the concentration we obtained for adult (5 months old WT) brain tissue lysate, representing adult physiological levels (*Figure 6B–C*). An increase of AICD in mature neurons has been reported in human AD brains and AD mouse models (*Ghosal et al., 2009*; *Lauritzen et al., 2012*). Using the same method, we therefore quantified AICD levels in the Tg2576 AD mouse model, at an early symptomatic age (5 months old) (*Figure 6B–C*, *Supplementary file 1*, one-way ANOVA). We could clearly observe an approximately fivefold increase of AICD in Tg2576 brains, when comparing to the adult physiological controls. Importantly, this quantitative analysis demonstrates that the addition of TAT-AICD-NLS to endogenous AICD levels in mature neurons (adult physiological AICD levels + TAT-AICD-NLS levels) could reach an AICD concentration in the range of values observed in the Tg2576 AD mouse model. Together, these data strengthen the hypothesis that an increased AICD level contributes to the synaptic deficits observed in AD.

## AICD perturbs synaptic signal integration and discharge probability by affecting the NMDAR, and consequently over-activating SK channels

To further explore our hypothesis that AICD could play a role in the synaptic deficits observed in AD, we investigated whether an increase of AICD in mature neurons could affect the way neurons integrate their synaptic inputs. Since the effect of AICD on synaptic GluN2B-NMDAR currents obtained with both exogenous delivery or in vivo viral-mediated expression was similar (*Figure 3*), we decided to perform these experiments by using in vivo transduction of AICD or AICD-NLS or GFP viruses to allow neurons to remain in their natural environment during AICD expression and to avoid long pre-incubation periods before electrophysiological recordings. We challenged these neurons at different frequencies of stimulation. EPSPs triggered by 5 pulses at different frequencies (0.1, 1, 10 and 50 Hz) were recorded in GFP, AICD and AICD-NLS neurons. For the first three frequencies, we measured EPSP amplitude for each pulse. At 50 Hz, during which the neuron will discharge action potentials (APs), we measured the discharge probability. GFP, AICD and AICD-NLS neurons exhibited similar mean EPSP amplitudes at low frequencies (0.1 and 1 Hz) (*Figure 7A and B*, *Supplementary file 1*; two-way ANOVA multiple comparisons). At 10 Hz, however, AICD and AICD-

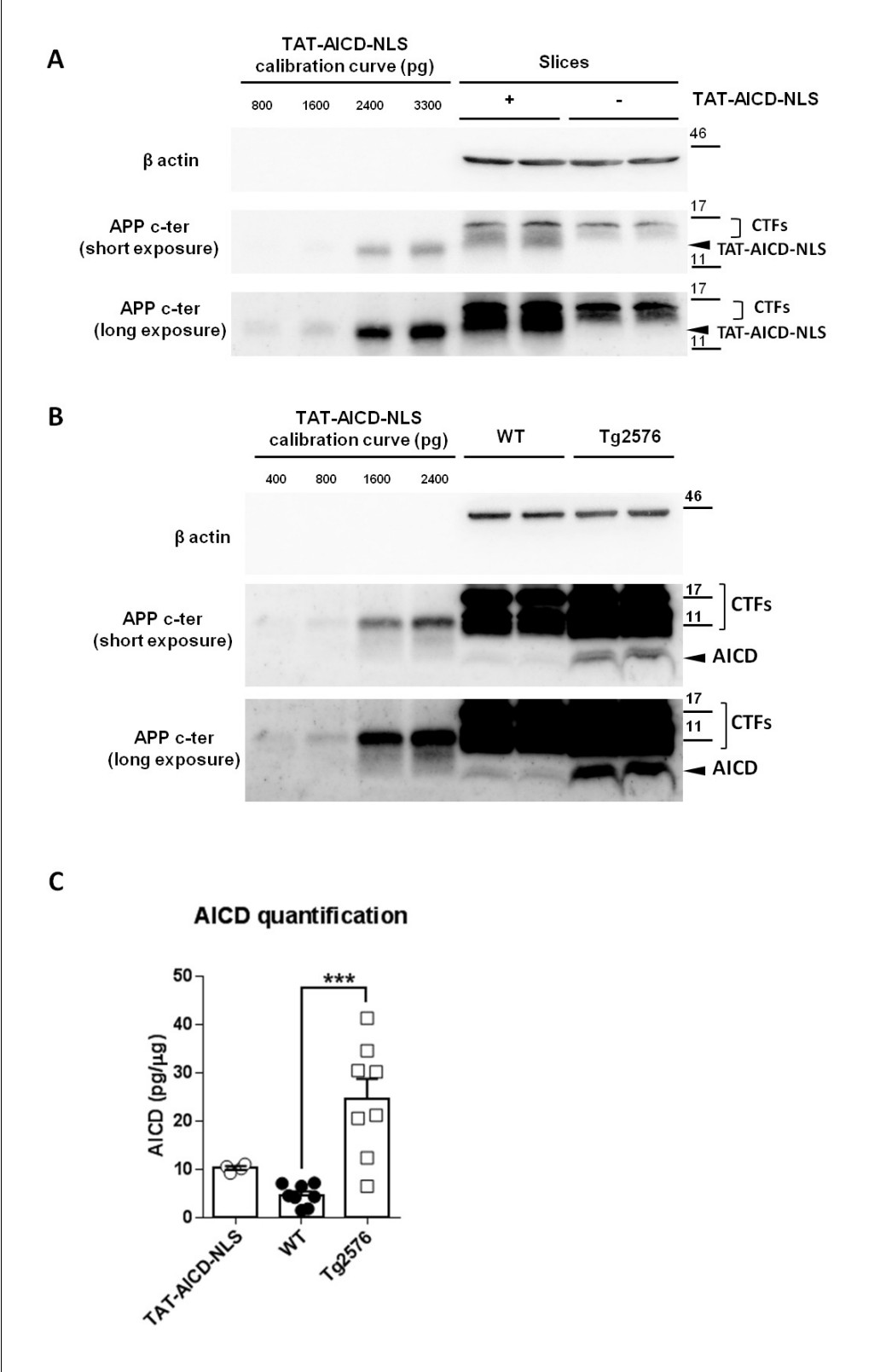

**Figure 6.** Physiological and pathological levels of AICD in brain. (**A**) Representative immunoblot of two samples of hippocampal slices with or without two hours of TAT-AICD-NLS (100 nM) processed for APP C-terminal detection (Y188 antibody) and actin (as loading control). Short and long exposures for Y188 are shown. (**B**) Representative immunoblot of two samples of lysates of cortices from 5 months old wild type control (WT) mice (physiological adult AICD levels) and APP transgenic littermates (Tg2576, pathological AICD levels) processed for APP C-terminal detection (Y188 antibody) and actin

*Figure 6 continued on next page*

Figure 6 continued

(as loading control). Short and long exposures for Y188 are shown. (C) After normalization to actin, AICD mass was quantified (pg/µg of tissue lysate) relative to the TAT-AICD-NLS calibration curve. N = 4 for hippocampal slices pre-incubated with TAT-AICD-NLS; N = 8 for WT Control brains; N = 8 for Tg2576 brains. ***p<0.001. Full statistical analysis is detailed in *Supplementary file 1*.

NLS neurons exhibited a decrease in mean EPSP amplitude after the second pulse compared to GFP neurons (*Figure 7C*, *Supplementary file 1*, two-way ANOVA multiple comparisons). Moreover, AICD and AICD-NLS neurons presented a significant decrease in AP discharge probability upon stimulation at 50 Hz (*Figure 7D*, *Supplementary file 1*; two-way ANOVA multiple comparisons). Intuitively, we did not expect that enhanced NMDAR function, observed in AICD neurons, could

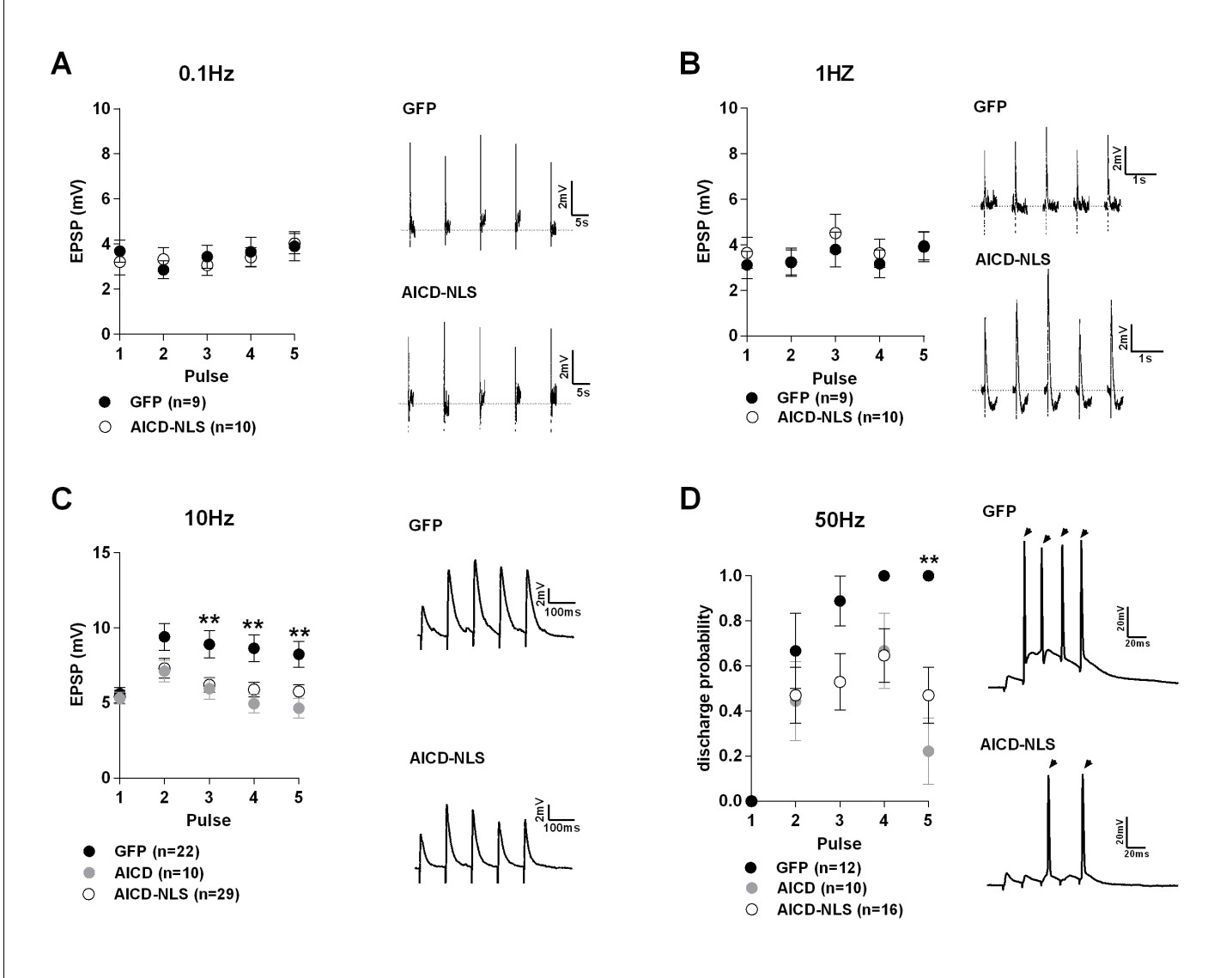

**Figure 7.** Synaptic signal integration and discharge probability are perturbed in AICD neurons. (A, B and C) Panels show absolute amplitude values of five consecutive EPSPs induced at the following frequencies: (A) 0.1 Hz, (B) 1 Hz, (C) 10 Hz. Traces show representative EPSCs recorded from GFP, AICD and AICD-NLS neurons. (D) Panel shows discharge probability of GFP, AICD and AICD-NLS neurons in response to a 50 Hz stimulation. Representative traces illustrating the failure in spike transfer in AICD-NLS neurons when compared to GFP neurons are shown. Arrows indicate the occurrence of action potentials. **p<0.01. Full statistical analysis is detailed in *Supplementary file 1*.

promote synaptic failure in a frequency-dependent manner. We, however, describe below how AICD-induced altered NMDARs currents affect the small-conductance $Ca^{2+}$-activated potassium channels (SK), which are at the origin of this failure.

In dendritic spines, NMDARs form an intimate $Ca^{2+}$-mediated feedback loop with SK2 (*Köhler et al., 1996*). Synaptic SK2 channels and NMDARs colocalize within 25–50 nm in the post-synaptic densities of CA1 pyramidal neurons. This SK2-NMDAR coupling allows for a tight negative regulation of NMDAR effects in spines, favoring rapid $Mg^{2+}$ reblocking of NMDARs, thus limiting the amplitude of synaptic potentials and reducing $Ca^{2+}$ transients, especially for stimulation frequencies higher than 5 Hz (*Figure 8A*) (*Stackman et al., 2002*; *Mateos-Aparicio et al., 2014*; *Hammond et al., 2006*). As NMDAR function is strongly enhanced in AICD neurons, we hypothesized that this could over-activate SK channels. We tested the effect of apamin, a specific antagonist of SK channels, on the amplitude of EPSPs in both GFP and AICD-NLS neurons. As illustrated in *Figure 8B and C*, apamin (100 nM) increased the mean amplitude of EPSPs in GFP neurons, as previously reported by others at this synapse (*Ngo-Anh et al., 2005*; *Hammond et al., 2006*). Apamin-induced potentiation of neurons was significantly stronger in AICD-NLS than in GFP neurons (*Figure 8B and C*; *Supplementary file 1*; two-way ANOVA). Moreover, blocking GluN2B with ifenprodil (5 μM) completely abolished this effect (*Figure 8B and C*; *Supplementary file 1*; two-way ANOVA). We thus evidenced over-activation of synaptic SK channels in AICD neurons due to excessive presence of GluN2B-NMDARs. We next asked if we could normalize the signal integration at 10 Hz frequency and above by blocking SK channels or GluN2B-NMDARs. As illustrated (*Figure 8D–G*; *Supplementary file 1*, two-way ANOVA and multiple comparisons), these phenotypes were rescued by partial blockade of GluN2B-NMDARs with ifenprodil (300 nM) or blockade of SK channels with apamin (100 nM). Together, these results show that increasing AICD production alters synaptic signal integration and AP discharge probability at 10 Hz frequencies and above, impairment due to excessive GluN2B-NMDARs and SK channel functions.

## LTP, but not LTD, is impaired in AICD neurons, a phenotype rescued by partial blockade of GluN2B subunits

The AICD-dependent synaptic signal integration and discharge probability alterations are likely to strongly perturb NMDAR-dependent synaptic plasticity, especially LTP, which is evoked by high frequency stimulation. As expected, long-term depression (LTD), induced by low-frequency stimulation (1 Hz), was equivalent in GFP, AICD and AICD-NLS neurons (*Figure 9A,B*, *Supplementary file 1*; one-way ANOVA), confirming that synapses of AICD neurons adequately respond to low-frequency inputs. By contrast, AICD and AICD-NLS neurons could not maintain LTP induced by high-frequency stimulation (100 Hz; *Figure 9C,D*, *Supplementary file 1*; one-way ANOVA), confirming that these synapses do not adequately respond to high-frequency inputs. Additional analysis of EPSC slopes during LTD and LTP induction in GFP and AICD-NLS neurons confirmed this finding. While AICD-NLS and GFP neurons displayed similar EPSC slopes during LTD induction, we could observe a strong decrease of EPSC slopes in AICD-NLS neurons during LTP induction (*Figure 9—figure supplement 1*, *Supplementary file 1*; two-way ANOVA).

To unequivocally link this AICD-dependent LTP impairment to enhanced synaptic GluN2B signaling, we asked if we could normalize LTP in AICD-NLS neurons by partially blocking GluN2B. We performed a dose-response analysis of ifenprodil application on LTP in GFP and AICD-NLS neurons. As expected (*Foster et al., 2010*), ifenprodil dose-dependently inhibited LTP in GFP neurons, with a maximal efficacy observed within the micromolar range (*Figure 9F*, *Figure 9—figure supplement 2* and *Supplementary file 1* ; two-way ANOVA). By contrast, a bell-shape effect was observed in AICD-NLS neurons and 300 nM ifenprodil fully rescued LTP (*Figure 9E,F*, *Figure 9—figure supplement 2* and *Supplementary file 1* ; two-way ANOVA). 300 nM ifenprodil also normalized EPSC slope amplitude during LTP induction (*Figure 9—figure supplement 1C,D* and *Supplementary file 1* ; two-way ANOVA). Therefore, AICD-dependent synaptic alterations spare LTD but strongly perturb LTP, a phenotype that is normalized by partial GluN2B antagonism.

## Discussion

Here, we demonstrate a critical and physiological role for AICD on controlling GluN2B-NMDARs at CA1 excitatory synapses. In addition, we provide strong evidence that an increase of AICD in mature

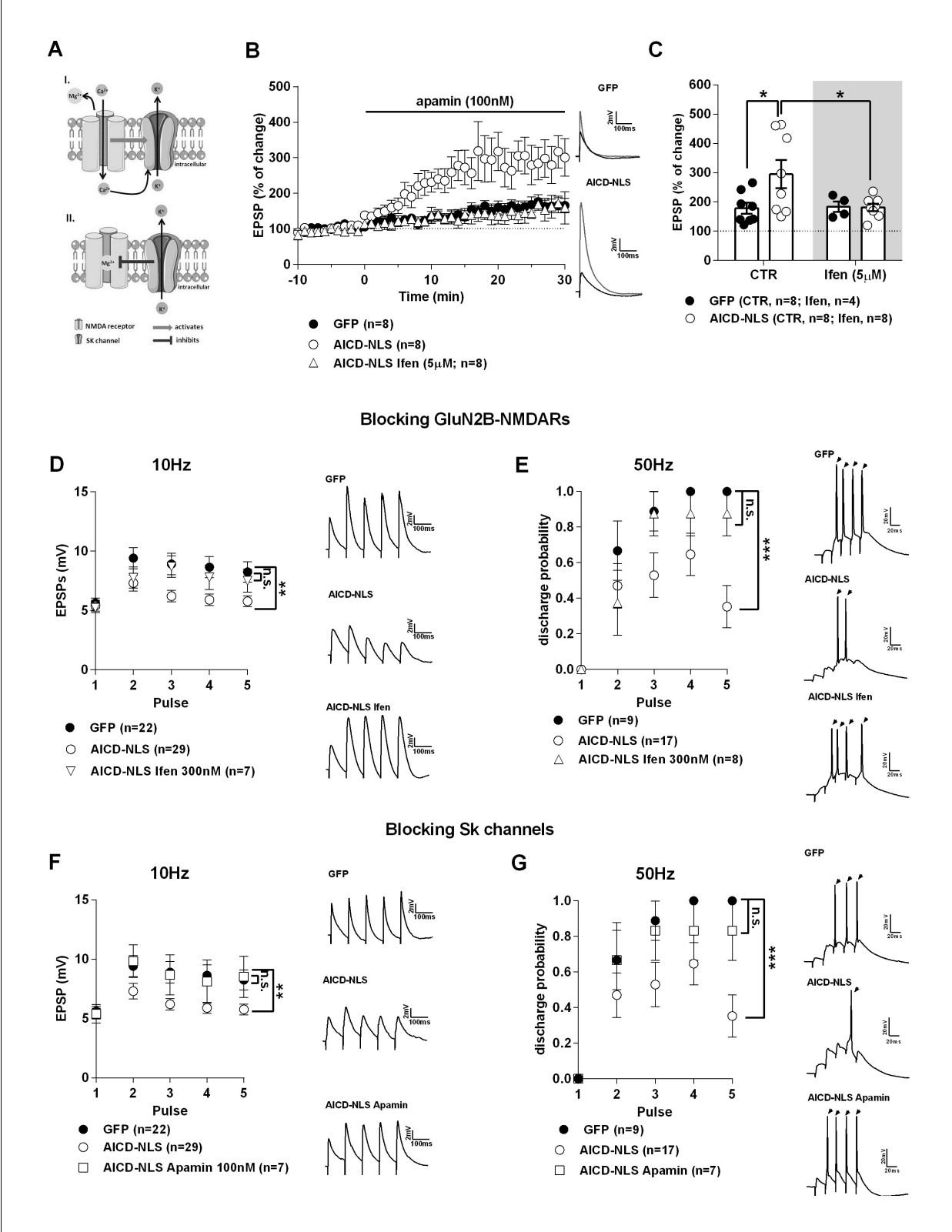

**Figure 8.** AICD perturbs synaptic signal integration and discharge probability by affecting the NMDAR-SK channel coupling. (**A**) Scheme of the NMDAR-SK channel coupling. Briefly, (I) When activated, NMDAR are permeable to $Ca^{2+}$, leading to its intracellular increase. This $Ca^{2+}$ activates the close-by SK channels, which open and allow the exit of $K^+$. (II) This hyperpolarizes the membrane and NMDAR become blocked again by $Mg^{2+}$. (**B**) Time course of apamin (100 nM) effect on EPSP amplitude recorded from GFP or AICD-NLS neurons in control conditions or from AICD-NLS neurons in

*Figure 8 continued*

presence of Ifenprodil (Ifen; 5 µM). Traces show the average of 30 EPSPs acquired before (black traces) and after apamin application (grey traces) in a GFP neuron and in an AICD-NLS neuron. (C) Scatter dot plot graph represents changes in EPSP amplitude in GFP and AICD-NLS neurons normalized to baseline values (100%, before apamin wash-in) in absence (CTR) or in presence of Ifenprodil (Ifen; 5 µM). (D, E and F, G) Panels show rescue of signal integration (10 Hz) and discharge probability (50 Hz) by partial blockade of GluN2B subunits by ifenprodil (300 nM) or blockade of SK channels by apamin (100 nM), as indicated. Representative traces illustrating the failure and rescue in spike transfer in AICD-NLS neurons when compared to GFP neurons are shown. Arrows indicate the occurrence of action potentials. Each dot in C represents one single neuron. *p<0.05; **p<0.01; ***p<0.001. Full statistical analysis is detailed in *Supplementary file 1*.

neurons, similar to levels observed in AD mouse models, can modify their synaptic NMDAR profile to an immature-like phenotype, with increased contribution of GluN2B-NMDAR. This over-activates SK channels, disrupting integration of synaptic signals in a frequency-dependent manner, thus compromising LTP, but not L

The evidence that AICD alters the AMPAR/NMDAR ratio confirms our hypothesis that AICD regulates synaptic transmission at CA1 excitatory synapses. Moreover, in the electrophysiological measures where AICD and AICDN-NLS were compared, AICD displayed a similar phenotype, albeit sometimes less pronounced, than AICD-NLS and expelling AICD from the nucleus by tagging it with a NES, completely prevented AICD effects on glutamatergic transmission. We, therefore, conclude that AICD exerts its effect at synapses via a nuclear-dependent mechanism. Accordingly, there is evidence that AICD is stabilized by the nuclear adaptor protein Fe65, which enables AICD to translocate to the nucleus (*Kimberly et al., 2001*; *Cupers et al., 2001*). Furthermore, a single Tyr(682) to Gly (Y682G) mutation in the AICD construct was sufficient to prevent AICD-dependent synaptic actions. Since this conserved Tyr(682) residue is a docking site for several cytoplasmic partners, such as Fe65 (*Grimm et al., 2013*), one might postulate that the absence of AICDY682G action could be due to its inability to translocate to the nucleus. Interestingly, although the YENPTY motif is present in the three members of the APP family, APLP2-ICD, but not APLP1-ICD, could mimic the synaptic phenotype. These data further support the well-documented functional redundancy between APP and APLP2 (*Shariati and De Strooper, 2013*). Of note, APLP2 more readily co-localizes than APLP1 with APP in intracellular compartments such as endosomes (*Kaden et al., 2009*). Also, APP and APLP2, but not APLP1, harbor a phosphorylation-dependent Pin1-binding site, which is implicated in APP C-terminal conformation and in Fe65 binding (*van der Kant and Goldstein, 2015*; *Pastorino et al., 2006*). Together, these observations could explain why, although all three APP family members contain the YENPTY motif, only APLP2-ICD mimics the AMPAR/NMDAR ratio phenotype. Interestingly, with regards to the experiments performed on *in utero* electroporated mice, the lack of AICD in APP KD neurons was not compensated by endogenous APLP2. Recently, Callahan et al. (*Callahan et al., 2017*) generated and characterized an APP conditional knockout mice (APP-flox), where deletion of APP is tamoxifen-inducible, to allow for clarification of the discrepancy in results observed between shRNA-induced APP reduction and germline APP knockout data. In accordance with our data, these authors could show that knocking down APP during a particular developmental window does not permit compensation by other APP family members, thus uncovering APP physiological roles that are otherwise masked by compensation in germline APP KOs.

As the vast majority of studies on APP have focused on its role for AD pathogenesis, its physiological function still remains elusive. Using neuron cultures, two studies had suggested a functional relationship between APP and NMDARs (*Cousins et al., 2009*) (*Hoe et al., 2009*). Also, the developmental role of APP in mammalian development persists after birth, as APP expression levels peak in the second post-natal week (*Löffler and Huber, 1992*). Strikingly, it coincides with the timing of the NMDAR developmental switch (*Williams et al., 1993*), suggesting that APP might play a role in this process. To our knowledge, our study is the first in vivo demonstration showing a crucial physiological role of APP in controlling synaptic GluN2B-NMDARs and AICD as the mediator of this control. This finding is in apparent disagreement with the observation reported by *Hoe et al. (2009)* where addition of the β-secretase-derived βAPP fragment (C99), precursor of Aβ and AICD (*Wolfe et al., 1999*), decreased NMDAR currents. These opposite effects may be due to concomitant generation of Aβ, which was shown to decrease synaptic NMDARs in cultured neurons (*Snyder et al., 2005*). Moreover, *Hoe et al. (2009)* suggested that APP enhanced GluN2B-NMDARs by decreasing

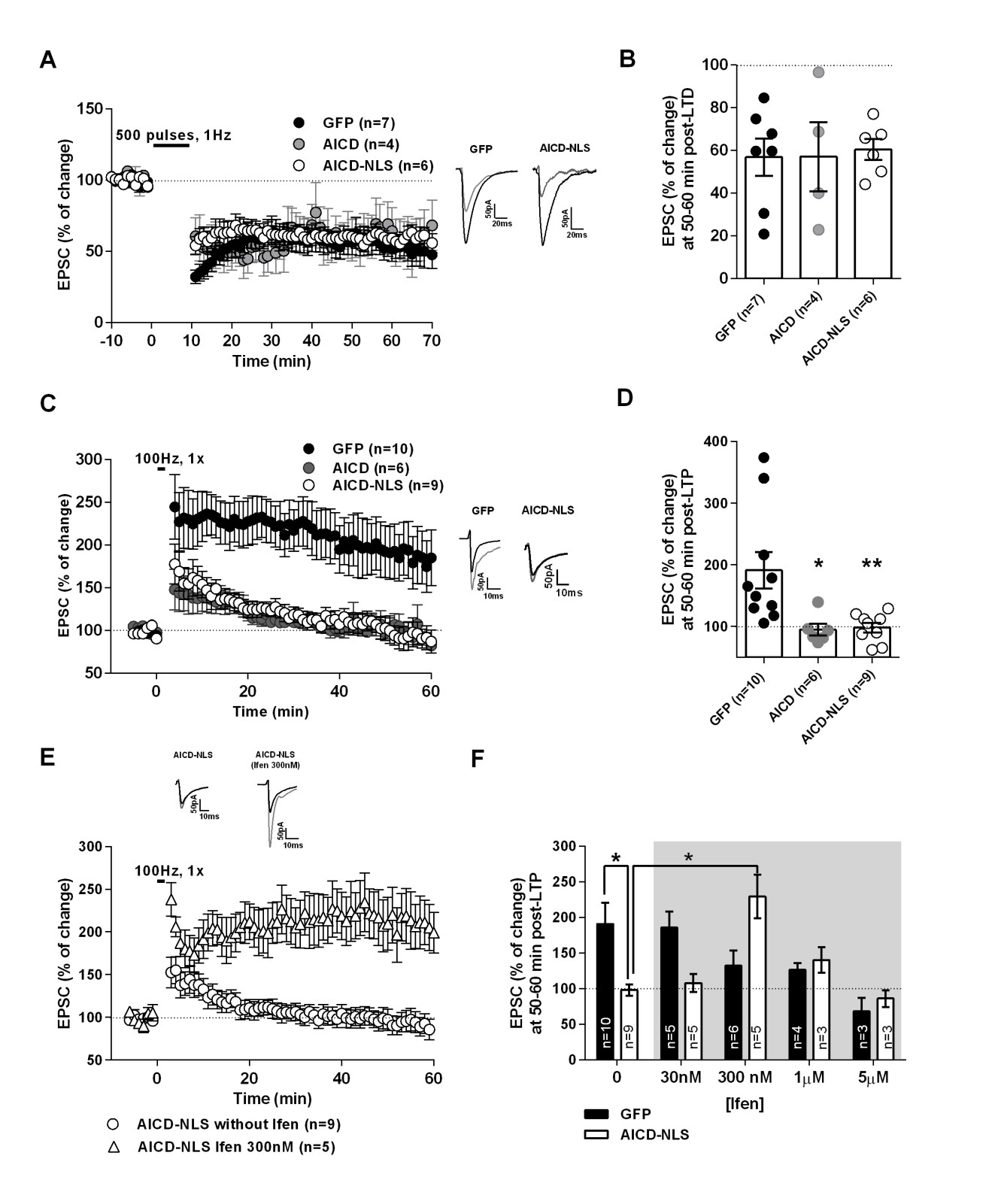

**Figure 9.** Increased AICD production spares LTD but impairs LTP, an effect rescued by partial blockade of GluN2B. (**A**) Summary graphs of EPSC amplitude pre- and post-LTD induction (−45 mV, 500 pulses, 1 Hz), elicited in GFP, AICD and AICD-NLS neurons. Sample EPSC traces pre-LTD (black) and 50–60 min post-LTD (grey) induction are shown. (**B**) Average LTD magnitude (at 50–60 min post-LTD induction) normalized to pre-LTD baseline values (100%) in GFP, AICD and AICD-NLS neurons. (**C**) Summary graphs of EPSC amplitude pre- and post-LTP induction (−10 mV, 100 Hz, 1 s) elicited

*Figure 9 continued on next page*

*Figure 9 continued*

in GFP, AICD and AICD-NLS neurons. Sample EPSC traces pre-LTP (black) and 50–60 min post-LTP (grey) induction are shown. (D) Average LTP magnitude (at 50–60 min post-LTP induction) normalized to pre-LTP baseline values (100%) in GFP, AICD and AICD-NLS neurons. (E) Panel shows summary graphs of EPSCs amplitude pre- and post-LTP induction (−10 mV, 100 Hz, 1 s) elicited in AICD-NLS neurons with or without 300 nM ifenprodil in vitro bath application. (F) Panel illustrates average LTP magnitude (at 50–60 min post-LTP induction) normalized to pre-LTP baseline values (100%) during 0 to 5 µM in vitro bath application of ifenprodil (Ifen). Each dot in (B) and (D) represents one neuron. *p<0.05; **p<0.01. Full statistical analysis is detailed in *Supplementary file 1*.

The following figure supplements are available for figure 9:

**Figure supplement 1.** AICD and GFP neurons exhibit different EPSC response profiles during LTP induction, but not during LTD induction.

**Figure supplement 2.** Ifenprodil affects LTP in a concentration-dependent manner, which is altered in neurons expressing AICD-NLS.

receptor internalization, but the authors did not clarify the mechanism. Also, it was shown that the APP intracellular domain associated-1 protein (AIDA-1) is implicated in endoplasmic reticulum export of GluN2B-containing NMDARs to synapses (*Tindi et al., 2015*) and that over-expression of APP resulted in enhanced cell surface NMDAR expression, which was not GluN2-subunit specific (*Cousins et al., 2009*), an effect that was also observed with over-expression of APLP1 and APLP2 (*Cousins et al., 2015*), suggesting an implication of the highly conserved intracellular domain in the control of NMDAR function. Yet, none of these studies showed if the APP/NMDAR interaction is direct or mediated via an intermediary protein and/or cascade of events. The assembly of our results supports a transcription-dependent mechanism, as AICD's effect on synaptic GluN2B-NMDAR currents was not observed if AICD was coupled to a nuclear export signal and was prevented by both mRNA translation and gene transcription inhibitors. Importantly, an increase in AICD levels correlated with an increase of GluN2B mRNA levels.

Several human studies have linked APP over-expression and alterations in APP processing to synaptic dysfunction and/or cognitive deficits. This is observed in trisomy 21 (Down syndrome) in humans (*McCarron et al., 2014*), and trisomy 16 in mice (*Cataldo et al., 2003*), where an additional copy of the chromosome harboring APP is found. It is also true for rare familial autosomal dominant AD cases (*Rovelet-Lecrux et al., 2006*) and de novo genetic variations (*Rovelet-Lecrux et al., 2015*), where duplication of the APP locus has occurred. In these contexts, APP over-expression will lead to increased production of the different proteolytic products, including AICD. Also, autosomal dominant genetic mutations observed in familial AD cases (e.g. KM670/671 NL Swedish mutation, A673V mutation; E682K Leuven mutation) and susceptibility loci discovered through genome-wide association studies (e.g. ATXN1) point to increased processing of APP via the amyloidogenic pathway (*Citron et al., 1992*; *Di Fede et al., 2009*; *Zhou et al., 2011*; *Bertram et al., 2010*), leading to an increase of APP products. Accordingly, besides A$\beta$, AICD and other APP fragments accumulate in brains from individuals with Trisomy 21 (*Sun et al., 2006*), in brains of humans diagnosed with AD (*Willem et al., 2015*; *Ghosal et al., 2009*) and in APP over-expressing AD mouse models (*Willem et al., 2015*; *Lauritzen et al., 2012*). It is therefore important to discriminate the contributions of the different APP fragments to the synaptic dysfunction observed in these scenarios. AICD has generally not been considered to participate in AD-associated synaptic failure. This is partly due to the lack of strong evidence demonstrating accumulation of AICD in human brains. However, it is well known that AICD is difficult to detect biochemically and unstable (*Hébert et al., 2006*; *Giliberto et al., 2010*; *Waldron et al., 2008*) and accumulation of this peptide is unlikely to occur even in disease conditions. Yet, according to our data, low nanomolar concentrations of AICD are sufficient to alter levels of synaptic GluN2B-NMDARs. We thus suggest that AICD does not need to accumulate to disrupt synapse function under pathological conditions. Notably, AICD could participate in excessive NMDAR function observed in APP over-expressing and presenilin mutant mouse models of AD (*Marchetti and Marie, 2011*; *Kaczorowski et al., 2011*; *Lanté et al., 2015*). An AICD-dependant mechanism could also be the target of the NMDAR-GluN2B antagonist memantine (*Xia et al., 2010*) commonly used for more than a decade to prevent declines in cognition, behavior and autonomy in AD patients, albeit with modest efficacy (*Matsunaga et al., 2015*).

Our findings clearly support a crucial physiological role of AICD in the control of NMDAR function at synapses. Moreover, we show that low nanomolar concentrations of AICD in mature neurons are sufficient to disrupt synapse signaling, thus raising the likelihood that AICD contributes to the AD synaptic failure, together with other APP fragments.

## Materials and methods

### Animals

For most experiments, sprague Dawley rats (RRID:RGD_5508397) (Janvier Labs, France) were used at post-natal days 21–30. For *in utero* electroporation, adult CD1 female mice (Janvier Labs, France) were used. For quantitative analysis of AICD levels in adult physiological and pathological conditions, 5-month old Tg2576 mice and WT littermates were used (Taconics, USA). All experiments were done according to policies on the care and use of laboratory animals of European Communities Council Directive (2010/63). The protocols we approved by the French Research Ministry following evaluation by a specialized ethics committee (protocol number 00973.02). All efforts were made to minimize animal suffering and reduce the number of animals used. The animals were housed three per cage under controlled laboratory conditions with a 12 hr dark light cycle, a temperature of 22 ± 2°C. Rats had free access to standard rodent diet and tap water.

Virus cloning c-DNA encoding a methionine followed by the last 50 amino acids of human $\beta$APP without or with the SV40 nuclear localization signal (NLS; PKKKRKV) or the HIV-1 Rev nuclear export signal (NES: LPPLERLTL) attached to its C-terminal end, called AICD and AICD-NLS and AICD-NES respectively, were inserted into the multiple cloning site (MCS) of the pAAV-IRES-hrGFP vector (Agilent Technologies, USA) by standard PCR cloning. The last 50 amino acids of human APLP1 and APLP2, called APLP1-ICD and APLP2-ICD respectively, were cloned using an identical procedure. The AICDY682G virus was created by site- directed mutagenesis of the AAV-AICD-IRES-GFP construct using the QuickChange II XL Site-directed mutagenesis kit (Agilent technologies, USA). Empty pAAV-IRES-hrGFP vector expressing only GFP was used as control (referred to as GFP virus). synGFP and synAICD viruses were constructed using the backbone vector pAAV-syn-GFP (Addgene plasmid #58867). In this vector, the GFP cDNA was replaced by either the hrGFP or the AICD-IRES-hrGFP using standard PCR cloning.

### Virus production

GFP, AICD, AICD-NLS, AICD-NES, AICDY682G, APLP1-ICD and APLP2-ICD synGFP and synAICD recombinant AAV virions, all expressing hrGFP, were produced essentially as described previously (*Hauck et al., 2003*).

Briefly, to produce rAAV virions containing 1:1 ratio of type 1 and type two capsid proteins, AAV-293 cells (RRID:CVCL_6871) (Agilent Technologies, USA) were co-transfected with the GFP, AICD, AICD-NLS, AICDY682G, AICD-NES, APLP1-ICD, APLP2-ICD, synGFP or synAICD plasmid and three helper plasmids (pH21, pRV1 and pFΔ6) using the calcium phosphate method. 65 hr post-transfection, the cells were harvested, rAAVs were purified using 1 ml HiTrap heparin columns (Sigma-Aldrich, France) and concentrated using Amicon Ultra centrifugal filter devices (Millipore, USA). The concentrated viral solution was aliquoted and stored at −80°C until further use. Titer quantification was performed in HT1080 cells as described by Agilent Technologies. Briefly, the number of infections rAAV particles was calculated by transducing HT1080 cells with serial dilutions of the viruses and counting GFP-positive cells. We consistently obtained titers of ~5×$10^8$ particles/ml for viruses for viruses exhibiting low transduction efficiency (GFP, AICD, AICD-NLS, AICDY682G, APLP1-ICD, APLP2-ICD) and ~5×$10^{12}$ particles/ml for viruses with high transduction efficiency (synGFP and synAICD).

### In vivo AAV injections

PND21-22 Sprague Dawley male rats (50–70 g) (RRID:RGD_5508397) were fully anesthetized and viruses were delivered to the CA1 hippocampal area by stereotaxic injection according to standard procedures. Briefly, animals were anesthetized with a mixture of ketamine (0.2 mg/g body weight) and xylazine (0.04 mg/g body weight) by intraperitoneal injection. Local analgesia was increased by injecting 35–40 μl of lidocaine (diluted at 0.5%) at site of incision. After immobilization on a

stereotaxic instrument, a hole was drilled (1–2 mm diameter) at −4 mm posterior and 2.5 mm lateral to bregma for injection in the CA1 region of the hippocampus. Using a stainless steel canula at a depth of 2.6 mm, viral solution (1 µl) was injected with a Harvard Apparatus pump at a flow rate of 0.2 µl/min. Rats were sutured and Ketofen (diluted in the bottle of water, 5 mg/L) was administrated during the first 24 hr after surgery to ease post-operative pain.

## Quantitative PCR

*Single cell:* cytoplasms of single neurons transduced with either the GFP, AICD, or AICD-NLS virus were collected from hippocampal slices 12–20 days post in vivo infection following a protocol optimized from *Lovatt et al., 2015*. Briefly, slices were prepared as for electrophysiology experiments (see below). Large glass patch pipettes (~1 MΩ) were front filled with about 2 µl of a HEPES buffered solution (in mM: 140 NaCl,5.4 KCl, 1 CaCl2, 1 MgCl2, 16 D-glucose, 10 HEPES) also containing a RNase inhibitor (0.5 U/µl; RNasin Plus, Promega, France). After placing the electrode on the neuron membrane, the cell body was aspirated with negative pressure. Successful aspiration was monitored visually by GFP fluorescence. The pipette with aspirated cytoplasm was immediately placed into PCR tubes and its tip was broken into a solution containing 0.05% Triton and the RNase inhibitor. Tubes were snap-frozen into liquid nitrogen and placed at −80°C until further use. All cytoplasms to be compared within one experiment were thawed at the same time and prepared for reverse transcription using the ProtoScript II Reverse Transcriptase Kit (New England BioLabs, France). cDNAs were preamplified using the PreAmp Master Mix Reagent Kit (Fluidigm, Netherlands) according to the manufacturer's instruction, and performing 19 PCR cycles. Each reaction was thereafter cleaned up with 8 units of Exonuclease 1 (New England BioLabs, France) (30 min at 37°C, 15 min at 80°C). Immediately, gene expression was assayed on the BioMark HD system (Fluidigm, Netherlands) using a Flex Six gene expression IFC (Fluidigm, Netherlands). Each 3 µL of sample premix consisted of 1.35 µL undiluted pre-amplified cDNA, 0.15 µL of 20x Flex Six Delta Gene Sample Reagent (Fluidigm, Netherlands) and 1.5 µL of 2x Sso Fast Eva green mastermix with Low Rox (Bio-Rad, France). Each 3 µL assay premix consisted of 0.075 µL of 100 µM primers (final concentration 2.5 µM primers), 1.5 µL 2x Assay loading reagent (Fluidigm, Netherlands) and 1.35 µL of TE buffer (10 mM Tris pH 8, 0.1 mM EDTA). Samples and assays were mixed inside the chip using the HX IFC controller (Fluidigm, Netherlands). Thermal conditions for qPCR were: 25°C for 30 min, 70°C for 60 min, 95°C for 1 min followed by 30 cycles of 96°C for 5 s, and 60°C for 20 s plus melting curve analysis. Data were processed by automatic threshold for each assay, with derivative baseline correction using BioMark Real-Time PCR Analysis Software 4.1.3 (Fluidigm, The Netherlands), including careful examination of melting curves in order to eliminate putative unspecific products. The quality threshold was set at the default setting of 0.65. Single cell quantification is reported as 1/Ct and was plotted in a box and whisker plot. PCR primers used for this experiment were: (1) Human APP C-term domain (Forward 5'−3' : CATGGTGTGGTGGAGGTTGA ; Reverse 5'−3': TAGCCGTTCTGCTGCATCTT) and (2) hrGFP (Forward 5'−3' : AACATCCTGTTCGGCAACCA ; Reverse 5'−3': CTCAGGATGTCGAAGGC-GAA).

*Hippocampus tissue:* 15–20 days after in vivo viral injections of synGFP or synAICD, quantitative PCR on micro-dissected transduced hippocampal tissues was performed using a standard quantitative PCR protocol. Briefly, cDNA was obtained using the ProtoScript II Reverse Transcriptase Kit (New England BioLabs, France). cDNA was diluted 1/10 and mixed with primer dilutions (10 µL each; Table S1) and SYBR green master mix for LightCycler 480 (Roche, France). qPCR was run using the LightCycler 480 II PCR machine (Roche, France) and analyzed using the associated software. Thermal conditions for qPCR were: 95°C for 5 min, followed by 40 cycles of 95°C for 10 s, 60°C for 10 s and 72°C for 10 s, plus melting curve analysis. Quantification of PCR on hippocampi was reported in mean fold change by pooling two independent experiments with a two-step normalization procedure. First, average Ct values for technical replicates of samples were calculated. Then, the 2dCt for each sample was calculated with the same GFP sample (as 1) as reference for each experiment. The fold change value was then calculated by normalization with the geometrical mean of 2dCt from two housekeeping genes, *Rpl13A* and *Ywhaz*, which were reported as the most stable for rat hippocampal tissue (*Bonefeld et al., 2008*). Primers were as follows: *Grin2b*: Forward 5'−3': AACCAAGA-GAGCCGACTAGC ; Reverse 5'−3': ACACCAACCAGAACTTGGGG); *Rpl13A*: Forward 5'−3' : AAGATCCGCAGACGCAAGG ; Reverse 5'−3': CTGTAGGGCACCTCACGATG; *Ywhaz*: Forward 5'−3' : TTGAGCAGAAGACGGAAGGT; Reverse 5'−3': GAAGCATTGGGGATCAAGAA). To pool

the two independent experiments together for statistical analysis, we further normalized each experimental value within the experiment to the average fold change obtained for GFP (as 1) in that experiment.

## TAT-AICD-NLS and TAT-scrAICD-NLS

TAT-AICD-NLS and TAT-scrAICD-NLS were synthesized by PSL GmbH (Heidelberg, Germany). TAT-AICD-NLS contained the TAT sequence (YGRKKRRQRRR), conferring cell permeability, fused to the N-terminal of AICD: VMLKKKQYTSIHHGVVEVDAAVTPEERHLSKMQQNGYENPTYKFFEQMQN. The SV40 nuclear localization signal (NLS; PKKKRKV) was added to the AICD C-terminal for nuclear translocation. The TAT-scrAICD-NLS contained the same TAT and NLS sequences at each end of a scrambled version of the AICD sequence: VQGITQKMYHNQEGKFLNQKVNVKTHMQFEHETLVDSMKA YYRVEEPSPA. TAT-AICD-NES was synthesized by Biosynthesis (Lewisville, USA). It contained the same TAT and AICD sequences as TAT-AICD-NLS, but the NLS was replaced by the NES sequence (LPPLERLTL). The purity of peptides was above 90–95%.

## Immunocytochemistry on neuronal cultures and N2a-APPKO cells

To generate APP KO in Neuro-2a ATCC CCL-131 cells (RRID:CVCL_0470) (obtained from ATCC (Lot: 62278033), we used the CRISPR-Cas9 system described by Ran et al. (*Ran et al., 2013*) Targeting sequences were designed using the web-based tool CRISPR Design (http://crispr.mit.edu/). The following target sequences directed to the exon1 of the murine *App* gene were used: sgRNA1-mAPP-KO, GCAGCATCGTGATCCTGCGT; sgRNA2-mAPP-KO, ACGGTTCGGGCTCTGGAGGT. The DNA sequences were synthesized (Sigma-Aldrich, Deisenhofen) and separately introduced into the plasmid vector pSpCas9(BB)-2A-Puro V2.0 (PX459; gift from Feng Zhang; Addgene plasmid # 62988). The murine N2a cell line was maintained in Dulbecco's modified Eagle's medium (DMEM/F12)+ GlutaMAX (Life Technologies, Germany) with 10% (v/v) fetal calf serum (FCS; Sigma-Aldrich) and 100 U/ml penicillin, 100 µg/ml streptomycin supplemented with non-essential amino acids. All cells were tested mycoplasma free using PCR primers of the VenorGeM Kit (Myco-for, GGGAG-CAAACAGGATTAGATACCCT and Myco-rev, TGCACCATCTGTCACTCTGTTAACCTC obtained from Sigma-Aldrich, Deisenhofen). All cells were mycoplasma free. N2a cells were transfected with the recombinant plasmids using Lipofectamine 2000 (Life Technologies, Germany). Twenty-four hours post-transfection, cells were selected with 3 µg/ml puromycin for 3 days. Single cell clones were then cultured with normal culture medium, followed by screening for genetic modifications in *App* by PCR amplification and by direct sequencing (GATC-Biotech, Germany) with following primers: mAPP_KO_for, ATCTTCCACTCGCACACGGA; mAPP_KO_rev2, ACGTCTCGAGATTCAAGCT (Sigma-Aldrich, Deisenhofen). The absence of protein was confirmed by Western blotting and immunocytochemistry. N2a-APPKO cells were incubated 1 hr, at 37°C 5% CO2, with 100 nM TAT-AICD-NLS or TAT-AICD-NES and were then processed for immunocytochemistry.

Primary hippocampal cultures were prepared from E18 pregnant Wistar rats (RRID:RGD_68115). Briefly, cells were plated in Neurobasal medium (Invitrogen) supplemented with 2% B27 (Invitrogen, France), 0.5 mM glutamine, 12.5 µM glutamate and penicillin/ streptomycin on 24 mm glass coverslips pre-coated with poly-L-lysine (0.1 mg/ml). Neurons (100,000 cells per coverslip) were then fed once a week in Neurobasal medium (Invitrogen, France) supplemented with 2% B27 (Invitrogen, France) and penicillin/ streptomycin until use. Hippocampal neurons (14–17 days in vitro (DIV)) were transfected using Lipofectamin 2000 (Invitrogen, France), according to the manufacturer's instructions, with 1 mg of plasmid shAPP active or inactive and 0,5 mg of pEGFP and used 16 hr post transfection. shAPP active and inactive cDNAs were generously provided by Dr Young-Pearse (Harvard Medical School, USA) (*Young-Pearse et al., 2007*).

N2a cells and primary neuronal cultures were fixed and permeabilized by fixation buffer A (paraformaldehyde 4%, Sucrose 0,147M, Triton 0,1%, PBS 1X) 2 min, following fixation buffer B, 58 min (paraformaldéhyde 4%, Sucrose 0,147M, PBS 1X). After washing 5 min in 1X PBS, cells were incubated 5 min in NH4Cl 50 mM, PBS 1X. Cells were rinsed by PBS 1 × 5 min following saturation while 45 min in saturation buffer (horse serum 10%, Triton 0,1%, PBS 1X). Rabbit anti-APP antibody (RRID: AB_873894) (Y188, Abcam, France) was then applied at 1:200 dilution for 2 hr at 4°C. Alexa 594-conjugated secondary antibody (RRID:AB_141637) (Donkey anti Rabbit, Lifetechnologies, France) was applied 1 hr at 1:200 dilution. For N2a cells, nuclei were stained with DAPI prior to mounting. Cells

were embedded in Mowiol. Images were acquired on Leica TSC/SP5 confocal microscope and processed with Image J software for quantification of APP staining in cell bodies or nucleus normalized to area.

## In utero electroporation of shAPP active and shAPP inactive

shAPP-active and shAPP-inactive cDNA were generously provided by Dr Young-Pearse (Harvard Medical School, USA) (*Young-Pearse et al., 2007*). A standard eGFP cDNA was used to mark electroporated neurons. *In utero* electroporation was performed at E14.5 as previously described (*Pacary and Guillemot, 2014*). Briefly, time-pregnant CD1 mice (RRID:IMSR_JAX:002962) were anesthetized with isoflurane and uteri were exposed through a 1 centimeter incision in the ventral peritoneum. Embryos were carefully lifted using ring forceps through the incision and placed on humidified gauze pads. DNA was prepared in endotoxin-free conditions (Qiagen, France) and each construct was injected at a final concentration of 1 µg/µl. Plasmid DNA solution mixed with 0.05% Fast Green (Sigma) was injected through the uterine wall into the telencephalic vesicle using pulled borosilicate needles (Harvard Apparatus, France) and a Femtojet microinjector (Eppendorf, France). Five electrical pulses were applied at 30V (50 ms duration) across the uterine wall at 1 s intervals using 5 mm platinum electrodes (Tweezertrode 45–0489, BTX, Harvard Apparatus, France) connected to an electroporator (ECM830, BTX). The hippocampus was targeted by orientating the electrodes as previously depicted (*Pacary and Guillemot, 2014*). The uterine horns were then replaced in the abdominal cavity and the abdomen wall and skin were sutured using surgical needle and thread. Pregnant mice were warmed on heating pad until they woke up. The whole procedure was complete within 20 min. Pups born after this procedure, were used for immunohistochemistry or electrophysiological experiments at PND 7–9.

## Immunofluorescent staining of shAPP active and inactive in utero electroporated brains and quantification analysis by confocal microscopy

Electroporated brains were removed from anesthetized pups at P7 subjected to intracardiac perfusion of PBS followed by 4% PFA in PBS. Brains were then post-fixed in 4% PFA overnight. PFA-fixed brains were washed in PBS and vibratome sectioned (40 µm). Sections were incubated in blocking buffer (10% goat serum; 0.1% Triton X-100; 0.5% BSA in PBS) for 30 min at room temperature. Sections were then incubated in chicken anti-GFP (RRID:AB_300798) (1:1000, Abcam, France), and rabbit anti-APP antibody (RRID:AB_2056556) (1:500, Y188, Abcam, France for 24 hr at 4°C in PBS with 1% goat serum, 0.1% Triton X-100% and 0.5% BSA. After two washes in PBS, sections were incubated with anti-chicken Alexa488 (RRID:AB_2534096) (1:1000) and anti-rabbit Alexa647 (1:200)-conjugated secondary antibodies (RRID:AB_2536183) (Invitrogen, France) for 2 hr at room temperature. After three washes in 0.1M Phosphate buffer, sections were mounted on glass slides using Mowiol. Images were acquired using a confocal microscope TCS SP5 (Leica Microsystems, France). Quantification analysis of APP staining, normalized to area, was performed using GFP staining to delineate cell body in Image J.

## Immunoblotting

N2a-KO cells and hippocampal slices pre-incubated with TAT-AICD-NLS peptide were collected and proteins were extracted with RIPA buffer (20 mM Tris-HCl (pH 7.5), 150 mM NaCl, 1 mM EDTA, 1 mM EGTA,1% NP-40,1% sodium deoxycholate, 2.5 mM sodium pyrophosphate + protease inhibitor). Homogenates were centrifuged at 14000 g for 30 min.

Five-month old WT adult and Tg2576 brains were dissected and snap-frozen in liquid nitrogen. Proteins were extracted in A-EDTA buffer (10 mM HEPES-KOH (pH 7.9), KCl 10 mM, $MgCl_2$1.5 mM, EDTA 0.1 mM + protease inhibitors). Briefly, each half brain resuspended in 3 ml buffer A-EDTA, six strokes were done in a 30 ml Teflon /glass homogenizer and Igepal 0.3% final was added just before six additional strokes.

Protein concentration of the cleared lysate was estimated with the *BCA* Protein *Assay* Kit (Pierce, USA) using BSA standard protein. 5 x Laemmli buffer was added to the lysates. The samples were sonicated for 10 min and boiled for 10 min. 20% of protein were loaded on 15% Tris-Glycine gel for N2a cell samples and on 16.5% Tris-Tricine gel for hippocampal samples and WT and Tg2576 brains,

transferred on nitrocellulose membrane after protein separation. The C-terminal domain of APP was detected with the rabbit monoclonal antibody Y188 (RRID:AB_2056556) (ab32136; Abcam, France; 1:1000 dilution) and $\beta$ actin was detected with the mouse monoclonal antibody AC-74 (RRID:AB_476743) (A5316; Sigma-Aldrich, France; 1:5000 dilution). The fragments were then detected with the respective HRP coupled anti-rabbit or anti-mouse secondary antibody (Sigma-Aldrich, France) followed by ECL (PerkinElmer, USA) detection with Fusion FX imager (Viber Lourmat, France). Quantification was done with ImageJ software (NIH, USA). The TAT-AICD-NLS dose-response calibration curve was obtained by quantifying the average value of chemiluminescence, obtained for at least two exposure times, relative to known doses of TAT-AICD-NLS, namely: 400 pg, 800 pg, 1600 pg and 2400 pg. To reduce the variability inter-experiments, we did a linear regression of all the values of chemiluminescence obtained for the known TAT-AICD-NLS doses at these two exposure times, which allowed to obtain the equation (y = 11.34 x). These known TAT-AICD-NLS doses were loaded in the same gel as the brain tissue lysate samples of interest. Each tissue lysate sample was normalized to its actin loading control. AICD levels in tissue lysates were calculated based on this equation using at least two exposure times per membrane.

## Electrophysiology

Acute transverse hippocampal slices (250 μm thick) were prepared 12–20 days after in vivo viral injections using standard procedures (*Marie et al., 2005*). Briefly, slices were cut on a vibratome (Microm HM600V, Thermo Scientific, France) in ice-cold dissecting solution containing (in mM): 234 sucrose, 2.5 KCl, 0.5 $CaCl_2$, 10 $MgCl_2$, 26 $NaHCO_3$, 1.25 $NaH_2PO_4$ and 11 D-glucose, oxygenated with 95% $O_2$ and 5% $CO_2$, pH 7.4. Slices were first incubated, for 60 min at 37°C, in an artificial CSF (aCSF) solution containing (in mM): 119 NaCl, 2.5 KCl, 1.25 $NaH_2PO_4$, 26 $NaHCO_3$, 1.3 $MgSO_4$, 2.5 $CaCl_2$ and 11 D-glucose, oxygenated with 95% $O_2$ and 5% $CO_2$, pH 7.4. Slices were used after recovering for another 30 min at room temperature. For all experiments, slices were perfused with the oxygenated aCSF at 31 ± 1°C in the continuous presence of 50 μM picrotoxin (Sigma-Aldrich, France, dissolved in DMSO) to block GABAergic transmission, except for experiments where LTP was investigated. In this case picrotoxin was not added to the aCSF solution to avoid the occurrence of spiking events. Recording pipettes (5–6 MΩ) for voltage-clamp experiments were filled with a solution containing the following: 117.5 mM Cs-gluconate, 15.5 mM CsCl, 10 mM TEACl, 8 mM NaCl, 10 HEPES, 0.25 mM EGTA, 4 mM MgATP and 0.3 NaGTP (pH 7.3; osmolarity 290–300 mOsm). For current-clamp experiments, the recording pipette solution contained (in mM): 135 gluconic acid (potassium salt: K-gluconate), 5 NaCl, 2 $MgCl_2$, 10 HEPES, 0.5 EGTA, 2 MgATP and 0.4 NaGTP (pH 7.25; osmolarity 280–290 mOsm). The Schaffer collateral pathway was stimulated at 0.10 Hz using electrodes (glass pipettes filled with aCSF) placed in the stratum radiatum.

Slices were visualized on an upright microscope with IR-DIC illumination and epi-fluorescence (Scientifica, Ltd, UK). Whole-cell recordings were performed using a Multiclamp 700B (Molecular Devices, UK) amplifier, under the control of pClamp10 software (RRID:SCR_011323) (Molecular Devices, UK).

After a tight seal (>1 GΩ) on the cell body of the selected neuron was obtained, whole-cell patch clamp configuration was established, and cells were left to stabilize for 2–3 min before recordings began. Holding current and series resistance were continuously monitored throughout the experiment, and if either of these two parameters varied by more than 20%, the experiment was discarded.

To calculate the AMPAR/NMDAR ratio (*Figure 2A*), cells were held at −65 mV to record EPSC_AMPAR and at +40 mV to record EPSC_NMDAR. EPSC_AMPAR amplitudes were calculated by averaging 30 consecutive EPSCs recorded at −65 mV and measuring the peak compared to the baseline. NMDAR EPSCs amplitudes were calculated by averaging 30 consecutive EPSCs recorded at +40 mV and measuring the amplitude 60 ms after EPSC onset compared to the baseline.

For AMPAR mEPSC recordings (*Figure 2B*), TTX (0.5 μM) was added to the external solution, and MgSO4 was lowered to 0.5 mM. At least 100 events were obtained from each cell. Recordings were analyzed using the clampfit software by applying a threshold search to detecting spontaneous events. The threshold for AMPAR mEPSCs detection was set at −5 pA for excluding electric noise contamination.

For comparing the amplitudes of EPSCs recorded from infected versus neighboring uninfected neurons, pairs of neurons were patched concomitantly (*Figure 2E–I*). Stimulus position and intensity

were set to evoke an EPSC of ~−100 pA in the infected neuron. We directly compared each pair of cells by normalizing the EPSC amplitude of the uninfected neuron to 100% (rather than comparing mean EPSC amplitudes; *Figure 2G and I*). When this procedure is used, each cell pair contributes equally to the calculated mean and the potentially distorting effects of individual EPSC measurements > 2 SD outside the mean are minimized.

Pharmacologically isolated $EPSC_{NMDAR}$ decay time, recorded from cells voltage clamped at +40 mV, was fitted with a double exponential function, using Clampfit software, to calculate both slow and fast decay time constants, $\tau_{fast}$ and $\tau_{slow}$, respectively (*Figure 3A*). The weighted time constant ($\tau_{weighted}$) was calculated using the relative contribution from each of these components, applying the formula: $\tau_w = [(a_f.\tau_f) + (a_s.\tau_s)]/(a_f + a_s)$, where $a_f$ and $a_s$ are the relative amplitudes of the two exponential components, and $\tau_f$ and $\tau_s$ are the corresponding time constants (*Figure 3A*). Different pharmacological tools were used: Ifenprodil (Sigma Aldrich, France) to selectively block GluN2B containing-NMDARs; Actinomycin (Sigma Aldrich, France) to inhibit gene transcription, Anisomycin (Sigma Aldrich, France) to inhibit mRNA translation and Apamin (Calbiochem, France) to block SK channels.

For measurement of EPSPs (*Figures 7* and *8*), synaptic potentials were recorded in whole-cell current-clamp mode. Subthreshold EPSPs were elicited by an electric stimulus of intensity that was approximately one-quarter of the stimulus required for evoking an action potential. To prevent epileptic discharges in the presence of picrotoxin, the CA3 region was microdissected before recordings. All recordings used cells with a resting membrane potential more negative than −62 mV and did not change by >2 mV during an experiment. Average resting membrane potentials were similar between neurons transduced with the different viruses (GFP, −68.67 ± 1.14; AICD-NLS, −68.30 ± 0.82).

To study the synaptic signal integration (*Figures 7* and *8*), we recorded EPSPs triggered by 5 pulses at different frequency stimulations (0.1, 1, 10 and 50 Hz; *Figure 7*). Stimulus intensity was adjusted to obtain the same amplitude of the first EPSP in the different infected neurons (~4–5 mV). No action potentials discharge was observed in the first three protocols. The amplitude of each EPSP was measured. At 50 Hz (*Figures 7D* and *8E and G*), during which neurons discharged APs, we measured discharge probability, considering the value of '1' whenever an AP was observed and '0' in absence of an AP.

To induce LTP, cells were held at −10 mV while stimulating afferent inputs with a HFS (100 pulses, 100 Hz) (*Figure 9C* to *Figure 9F*). To induce LTD, cells were held at −45 mV while stimulating afferent inputs at 1 Hz for 500 pulses (*Figure 9A–B*). Ifenprodil was applied at least 30 min before LTP induction.

Time course graphs (*Figures 3B, C, 5E, G, 8B, 9A, C and E, Figure 9—figure supplement 2*) were obtained by normalizing each experiment according to the average value of all points on the 5 or 10 min stable baseline, as indicated, before drug application or LTD/LTP induction. Drugs or LTD/LTP induction effects were calculated as percentage change in mean average amplitude of responses measured during the last 5 min of recording (35–40 min after drug perfusion or 50–60 min after LTD/LTP induction).

## Statistical analysis

Upon design of the major experiments for which the outcome could not be predicted (*Figure 2A and H*, *Figure 7*), we planned experiments on a minimum of 3 independent animals and 8 cells as per our previous expertise when running statistical analysis on these types of data. For all other experiments where the outcome could be predicted that is, existing previous literature already describing expected effect (e.g. ifenprodil effect in PND7 and PND30 in wild-type rodent) or if our own data within this study strongly suggested outcome (e.g. no perturbation of LTD but LTP impairment because of disrupted synaptic signal integration at high stimulation frequencies), we planned experiments on a minimum of 3 independent animals and 4 cells. On average, we replicated our findings in independent 5 animals and 10 cells per condition (see full numbers in *Supplementary file 1*). Results are shown as mean ± s.e.m. 'n' refers to the number of cells examined, unless otherwise stated. Statistical significance of differences between means was determined through Prism 6.0 software (GraphPad) (RRID:SCR_002798) and is presented in detail in *Supplementary file 1*, significance was taken at p<0.05. Wilcoxon matched pairs were performed to analyze significance between the simultaneous responses recorded from uninfected (considered

100%) and its neighbor infected neuron. Student's t-test and Mann Whitney test were performed whenever two conditions were compared, according to the samples distribution normality. One factor analysis of variance (ANOVA) was performed to analyze significance among the different conditions. In studies where drug effects were evaluated, two factor analysis of variance (ANOVA) was performed to detect interaction between treatment and cell type.

## Acknowledgements

We thank Stefan Kins (Technische Universität Kaiserslautern, Germany) for APLP1 and APLP2 cDNAs. pAAV-Syn-GFP was a gift from Edward Boyden (Addgene plasmid # 58867). We thank Feng Zhang (Broad Institute) for providing PX459V2.0. We thank Tracy Young-Pearse (Harvard Medical School, USA) for providing shAPP active and shAPP inactive cDNA constructs. We thank Eric Duplan (IPMC) for help with cloning. We thank Rainer Waldmann and Pascal Barbry (IPMC, France) for discussions on single cell qPCR analysis. We thank Sandra Ruiz-Garcia and Julien Pigeon (IPMC, France) for technical help. We thank the IPMC Functional Genomics platform for their services. We thank Jacques Barik (IPMC, France) and Corinne Beurrier (IBDM, France) for scientific discussions pertaining to this project and critical reading of the manuscript.

## Additional information

### Funding

| Funder | Grant reference number | Author |
|---|---|---|
| Centre National de la Recherche Scientifique | | Paula A Pousinha Carole Gwizdek Gihen Dhib |
| Fondation pour la Recherche Médicale | SPF20130526736 | Paula A Pousinha |
| Fondation Plan Alzheimer | Senior Innovative Grant 2010 | Xavier Mouska Elisabeth F Raymond Hélène Marie |
| Canceropôle PACA | | Laure-Emmanuelle Zaragosi |
| Agence Nationale de la Recherche | ANR-10-INBS-09-03 ANR-10-INBS-09-02 | Xavier Mouska Laure-Emmanuelle Zaragosi |

The funders had no role in study design, data collection and interpretation, or the decision to submit the work for publication.

### Author contributions

PAP, Conceptualization, Data curation, Formal analysis, Validation, Investigation, Visualization, Methodology, Writing—original draft, Project administration, Writing—review and editing, Designed the study and interpreted the results. Performed all in vivo stereotaxic surgeries, electrophysiological recordings and analysis, collected cytoplasms for single cell qPCR and prepared the PFA fixed the brains. Wrote the manuscript with input from the other authors; XM, Data curation, Methodology, Produced viruses, cloned viruses. Performed single cell and tissue quantitative PCR experiments. Performed biochemical analysis; EFR, Resources, Methodology, Produced viruses, cloned viruses; CGw, Resources, Data curation, Methodology, Performed biochemical analysis; GD, Data curation, Methodology, Performed Immunofluorescent staining and confocal microscopy and cell culture experiments and quantification; GP, Resources, Methodology, Produced hippocampal neuronal cultures; L-EZ, Data curation, Methodology, design and interpret single-cell and tissue quantitative PCR experiments; CGi, Methodology, Produced the APP deficient N2a cells; IB, Data curation, Methodology, Performed stereotaxic surgery to inject the synvirus; EP, Methodology, Writing—review and editing, Performed in utero electroporation and prepared the PFA fixed the brains. Review paper writing; MW, Methodology, Writing—review and editing, Produced the APP deficient N2a cells. Review paper writing; HM, Conceptualization, Supervision, Funding acquisition, Validation, Investigation, Visualization, Methodology, Writing—original draft, Project administration, Writing—review and editing, Designed the study and interpreted the results. Produced viruses, cloned viruses.

Performed single cell and tissue quantitative PCR experiments. Performed biochemical analysis. Wrote the manuscript with input from the other authors

### Author ORCIDs
Paula A Pousinha, http://orcid.org/0000-0002-5992-9418
Xavier Mouska, http://orcid.org/0000-0002-4263-543X
Hélène Marie, http://orcid.org/0000-0003-2310-6097

### Ethics
Animal experimentation: All experiments were done according to policies on the care and use of laboratory animals of European Communities Council Directive (2010/63). The protocols we approved by the French Research Ministry following evaluation by a specialized ethics committee (protocol number 00973.02). All efforts were made to minimize animal suffering and reduce the number of animals used.

## Additional files

### Supplementary files
• Supplementary file 1. Statistics. (A) AICD increases NMDAR, but not AMPAR, transmission in CA1 pyramidal neurons. (B) AICD regulates synaptic currents mediated by increasing synaptic GluN2B contribution. (C) Statistics (AICD regulates GluN2B mRNA levels). (D) APP knock down eliminates synaptic GluN2B NMDARs, an effect rescued by nuclear AICD delivery. (E) Physiological and pathological levels of AICD in the brain. (F) AICD perturbs synaptic signal integartion and discharge probability. (G) AICD perturbs synaptic signal integration and discharge probability by affecting the NMDAR-SK2 channel coupling. (H) LTP, but not LTD, is impaired in AICD neurons, a phenotype rescued by partial blockade of GluN2B subunits.

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
