## [Decision Letter]

Thank you for submitting your article "Physiological and pathophysiological control of GluN2B at synapses by the C-terminal domain of APP" for consideration by *eLife*. Your article has been reviewed by three peer reviewers, and the evaluation has been overseen by a Reviewing Editor and a Senior Editor. The following individual involved in review of your submission has agreed to reveal their identity: Ilya Bezprozvanny (Reviewer #1).

The reviewers have discussed the reviews with one another and the Reviewing Editor has drafted this decision to help you prepare a revised submission.

Summary:

In this study, Pousinha et al. investigated the role of APP intracellular domain (AICD) in regulating the basal transmission and long-term synaptic plasticity at the glutamatergic CA3-CA1 hippocampal synapses. The authors report that AICD specifically alters GluN2B-NMDAR transmission and NMDAR-dependent synaptic long-term potentiation. To this end, the authors used a combination of molecular tools to manipulate peptide expression in vivo and ex vivo and electrophysiological recordings. They further show that enhanced NR2B function leads to enhanced coupling with SK2 channels, which in turn causes disruption in synaptic signal integration. This is a very interesting study that tackles an important aspect of the APP functions at excitatory synapses. However, a biological role (or even presence of substantial amounts) remains elusive. The authors should provide additional evidence to support physiological relevance of their findings.

Essential revisions:

1) Physiological relevance:

The authors provide little, if any, evidence that the AICD peptide (not only the mRNA) is indeed present in the cytoplasmic and/or dendritic compartments, and that the AICD-NLS peptide is enriched within the nucleus. How is amount of AICD present in neurons under normal physiological conditions comparable to 30 nM of TAT-AICD-NLS peptide, the concentration altering NR2B receptor function (Figure 3).

Figure 4 shows that knockdown of APP by shAPP causes complete disappearance of ifenprodil-sensitive component (Figure 4).

a) Quantification should be provided for both, immunoblot and the immunocytochemical evidence.

b) It is expected that APLP2-ICD should compensate for the lack of AICD in APP KD neurons (based on Figure 2). Why this is not happening?

2) Mechanism of AICD effects:

Nuclear targeting is essential for effects on NMDA currents (Figure 1). This indicates that AICD acts as a transcription factor in these experiments. However, application of TAT-AICD-NLS exerted effect on NMDAR within 2 h of incubation. Is there enough time for TAT-AICD-NLS to act by gene expression mechanisms? Necessary control here is TAT-AICD-NES peptide. Additional control is to test effects of inhibitors of protein transcription and translation. Is synthesis of new proteins necessary for AICD effects on NMDARs?

The authors propose that AICD action starts at the transcriptional level. Accordingly, the transcription of GluN2B itself and/or the transcription of other genes involved in the expression and/or trafficking of GluN2B should be increased. The authors should at least study if the GluN2B expression levels increase in the presence of AICD by quantitative PCR measurements. They have used that approach at the single cell level to demonstrate the expression of AICD, therefore they have at their fingertips the opportunity to demonstrate if GluN2B levels are increased.

3) Relevance to Alzheimer's pathophysiology: The authors state that they observed increased AICD in neurons from Tg2576 mouse model (data not shown). Do the levels of AICD in AD neurons reach the concentrations needed to exert effect on NR2B receptors? One way to test this will be to perform Western blotting with slices incubated with 30 nM and 100 nM of TAT-AICD peptide and compare levels of AICD detected in these experiments with levels of AICD detected by Western blotting of samples from AD mouse model (or human AD brains). Are they comparable?

---

## [Author Response]

*Essential revisions:*

*1) Physiological relevance:*

*The authors provide little, if any, evidence that the AICD peptide (not only the mRNA) is indeed present in the cytoplasmic and/or dendritic compartments, and that the AICD-NLS peptide is enriched within the nucleus.*

For immunohistochemistry, the currently available commercial antibodies against the C-terminal domain of APP do not permit to distinguish between full-length APP, CTFs or AICD nor to distinguish between endogenous AICD and synthetic AICD peptides. In an effort to specifically analyze the localization of the synthetic AICD peptide, we tried the two commercially available antibodies against the TAT sequence that was introduced in the synthetic AICD peptide, but none recognized this TAT sequence successfully (data not shown). We thus decided to work in an APP knock-out background and to use the Y188 antibody recognizing the c-terminal domain APP. We used N2a cells that were KO for APP (see Figure 3). We first performed a localization study of our virally-encoded AICD and mutants. However, unfortunately, these N2a cells were not efficiently transduced by our AAV viruses expressing the different mutants of AICD (data not shown). These cells were however efficiently penetrated by the synthetic peptides expressing TAT. We thus could show that TAT-AICD-NLS is indeed enriched in the nucleus (see Figure 3). We compared this nucleus/cytoplasm ratio to the one obtained in N2a APP KO cells incubated in the newly synthesized TAT-AICD-NES, demonstrating that, in presence of NLS, there is an enrichment of this peptide in the nucleus, comparing to TAT-AICD-NES (Figure 3).

This new data has been described in the main text of the manuscript: “In a control experiment … compared to TAT-AICD-NES”.

*How is amount of AICD present in neurons under normal physiological conditions comparable to 30 nM of TAT-AICD-NLS peptide, the concentration altering NR2B receptor function (Figure 3).*

As suggested by the reviewers (see also point 3 of major revisions below), we now introduced a new section in the Results “Physiological and pathological levels of AICD in brain tissue”, dedicated to the analysis of AICD levels. We first quantified the TAT-AICD-NLS levels incorporated in cells after pre-incubation of hippocampal slices with TAT-AICD-NLS (100 nM) for two hours. We decided to use the 100 nM concentration for this quantification, because most of our experiments and controls have been performed at this concentration. This tissue was then processed for western-blotting analysis. In each experiment, the TAT-AICD-NLS mass was calculated based on a dose-response calibration curve of known doses of TAT-AICD-NLS (Figure 6) (see Materials and methods, subsection “Immunoblotting” for details). To allow a comparison between this result and the endogenous physiological and pathological AICD levels, we quantified the AICD levels in the Tg2576 AD mouse model brains, at an early symptomatic age (5 month old) (Figure 6), and their respective age-matched wild type littermates (WT, Figure 6). We could clearly observe an approximately fivefold increase of AICD in Tg2576 brains, when comparing to the adult physiological controls. Importantly, this quantitative analysis demonstrates that the addition of TAT-AICD-NLS to endogenous AICD levels in mature neurons (adult physiological AICD levels + TAT-AICD-NLS levels) could reach an AICD concentration in the range of values observed in the Tg2576 AD mouse model.

This new data has been described in the main text of the manuscript: “Our data… observed in AD” and a dedicated figure was added (Figure 6).

*Figure 4 shows that knockdown of APP by shAPP causes complete disappearance of ifenprodil-sensitive component (Figure 4). a) Quantification should be provided for both, immunoblot and the immunocytochemical evidence.*

We have now provided quantification of effective down-regulation of APP expression by the active shRNA against APP (shAPP active) in primary cultures of hippocampal neurons (see Figure 5) and in slices after in utero electroporation (see Figure 5) (see also subsection “APP knock-down eliminates synaptic GluN2B NMDARs, an effect rescued by nuclear AICD delivery”, first paragraph).

*b) It is expected that APLP2-ICD should compensate for the lack of AICD in APP KD neurons (based on Figure 2). Why this is not happening?*

This is an interesting observation. Recently, Callahan et al. (Callahan et al. 2017) generated and characterized an APP conditional knockout mice (APPflox), where deletion of APP is tamoxifen-inducible to allow for clarification of the discrepancy in results observed between shRNA-induced APP reduction and germline APP knockout data. In accordance with our data, these authors could show that knocking down APP during a particular developmental window does not permit compensation by other APP family members, thus uncovering APP physiological roles that are otherwise masked by compensation in germline APP KOs. We have now added this point to our Discussion and the reference to the main manuscript (Discussion, second paragraph).

*2) Mechanism of AICD effects:*

*Nuclear targeting is essential for effects on NMDA currents (Figure 1). This indicates that AICD acts as a transcription factor in these experiments. However, application of TAT-AICD-NLS exerted effect on NMDAR within 2 h of incubation. Is there enough time for TAT-AICD-NLS to act by gene expression mechanisms? Necessary control here is TAT-AICD-NES peptide. Additional control is to test effects of inhibitors of protein transcription and translation. Is synthesis of new proteins necessary for AICD effects on NMDARs?*

We have now added the results obtained with the newly synthesized TAT-AICD-NES peptide control. We observed that AICD, when coupled to NES, is unable to significantly increase synaptic GluN2B-NMDAR currents (Figure 3). We also tested the effects of the transcription inhibitor, actinomycin, and the translation inhibitor, anisomycin. Both prevented the actions of TAT-AICD-NLS peptide on GluN2B currents (see Figure 3), demonstrating that AICD exerts its effect via a gene transcription and protein translation dependent-mechanism.

This new data has been described in Figure 3 and the main text of the manuscript: “To confirm… is transcription-dependent”. It is also briefly discussed in the Discussion: “The assembly of … transcription inhibitors”.

*The authors propose that AICD action starts at the transcriptional level. Accordingly, the transcription of GluN2B itself and/or the transcription of other genes involved in the expression and/or trafficking of GluN2B should be increased. The authors should at least study if the GluN2B expression levels increase in the presence of AICD by quantitative PCR measurements. They have used that approach at the single cell level to demonstrate the expression of AICD, therefore they have at their fingertips the opportunity to demonstrate if GluN2B levels are increased.*

As suggested by the reviewers, we have used cytoplasms from neurons expressing GFP or AICD and GFP to quantify endogenous levels of GluN2B mRNA by single cell quantitative PCR. Unfortunately, while this approach worked well for over-expressed cDNA of GFP and AICD, the variability was too high for adequate quantification of this endogenous gene. As an alternative, we constructed new high titer AAVs encoding GFP or AICD together with GFP under the synapsin promoter (called synGFP and synAICD viruses; Figure 4) to direct specific expression of AICD cDNA (without NLS) in neurons with high transduction efficiency (Figure 4). We micro-dissected hippocampi regions highly transduced by synGFP or synAICD for quantitative PCR analysis. As expected, AICD expression was increased in SynAICD-expressing hippocampi compared to synGFP transduced tissue (Figure 4). Importantly, SynAICD-expressing hippocampi displayed a significant increase in endogenous GluN2B mRNA levels compared to synGFP-expressing hippocampi (Figure 4), demonstrating that increasing AICD levels correlates with increased Grin2b expression.

This new data constitutes a new Results section “AICD regulates GluN2B mRNA levels” and has been described in Figure 4 and in the main text of the manuscript: “Our data… Grin2b expression”. It is also briefly discussed in the Discussion: “Importantly,… GluN2B mRNA levels”. Full investigation of the underlying molecular mechanism linking nuclear AICD to Grin2b expression levels will be the subject of a future study.

*3) Relevance to Alzheimer's pathophysiology: The authors state that they observed increased AICD in neurons from Tg2576 mouse model (data not shown). Do the levels of AICD in AD neurons reach the concentrations needed to exert effect on NR2B receptors? One way to test this will be to perform Western blotting with slices incubated with 30 nM and 100 nM of TAT-AICD peptide and compare levels of AICD detected in these experiments with levels of AICD detected by Western blotting of samples from AD mouse model (or human AD brains). Are they comparable?*

As detailed above, we have now performed a quantification of AICD levels (pg/µg of tissue lysate) (see Figure 6) demonstrating that levels of AICD reached after pre-incubation of hippocampal slices with 100 nM TAT-AICD-NLS are comparable to levels observed in physiopathological levels (WT and Tg2576 brain levels).